# Beyond Reverse KL: Generalizing Direct Preference Optimization with Diverse Divergence Constraints

**Chaoqi Wang    Yibo Jiang    Chenghao Yang    Han Liu    Yuxin Chen**
Department of Computer Science, University of Chicago
Correspondence to `chaoqi@uchicago.edu`

## Abstract

The increasing capabilities of large language models (LLMs) raise opportunities for artificial general intelligence but concurrently amplify safety concerns, such as potential misuse of AI systems, necessitating effective AI alignment. Reinforcement Learning from Human Feedback (RLHF) has emerged as a promising pathway towards AI alignment but brings forth challenges due to its complexity and dependence on a separate reward model. Direct Preference Optimization (DPO) has been proposed as an alternative, and it remains equivalent to RLHF under the reverse KL regularization constraint. This paper presents $f$-DPO, a generalized approach to DPO by incorporating diverse divergence constraints. We show that under certain $f$-divergences, including Jensen-Shannon divergence, forward KL divergences and $\alpha$-divergences, the complex relationship between the reward and optimal policy can also be simplified by addressing the Karush–Kuhn–Tucker conditions. This eliminates the need for estimating the normalizing constant in the Bradley-Terry model and enables a tractable mapping between the reward function and the optimal policy. Our approach optimizes LLMs to align with human preferences in a more efficient and supervised manner under a broad set of divergence constraints. Empirically, adopting these divergences ensures a balance between alignment performance and generation diversity. Importantly, $f$-DPO outperforms PPO-based methods in divergence efficiency, and divergence constraints directly influence expected calibration error (ECE).

## 1 Introduction

The increasing capabilities of large language models (Bubeck et al., 2023; OpenAI, 2023) hold promise for achieving artificial general intelligence. However, they also pose safety concerns within the scope of AI risk (Amodei et al., 2016; Hendrycks et al., 2023). Some of the hazardous capabilities an AI system may possess include social manipulation (Hendrycks et al., 2023), AI-enabled cyberattacks (Shevlane et al., 2023), and enhanced pathogens (Urbina et al., 2022). These could be misused by humans or exploited by the AI system itself. Consequently, AI alignment research becomes critically important in ensuring AI systems are robustly aligned with human values.

Reinforcement Learning from Human Feedback (RLHF) has emerged as a concrete research agenda, proving effective in aligning model behaviors with human preferences and instruction following (Christiano et al., 2017; Bai et al., 2022a; Touvron et al., 2023). Given the challenge of specifying an objective that accurately represents human preferences in RLHF, researchers typically collect a dataset that reflects human preference in terms of model-wise generation comparisons (Bai et al., 2022a; LAION-AI, 2023). Subsequently, a reward model is trained under the Bradley-Terry model (Bradley & Terry, 1952) to infer the human's objective from the collected dataset. The language model is then fine-tuned using RL algorithms such as Proximal Policy Optimization (Schulman et al., 2017; Ouyang et al., 2022) or Advantage Actor-Critic (Mnih et al., 2016; Glaese et al., 2022) to maximize the reward. This process is carried out while ensuring that the model does not deviate significantly from its original form, using the reverse KL divergence penalty.

While effective, the RLHF pipeline is significantly more complex than supervised learning. In particular, RLHF necessitates training a separate reward model. The quality of this model ultimately determines the performance of reinforcement fine-tuning, and the language model may exploit errors present within the reward model (Gao et al., 2023). Additionally, RL algorithms, such as PPO (Schulman et al., 2017), are less stable and more memory-demanding than supervised learning (Touvron et al., 2023). These challenges pique interest in searching for alternatives to the RLHF pipeline. Such efforts include Reward rAnked FineTuning (RAFT) (Dong et al., 2023), Rank Responses to align Human Feedback (RRHF) (Yuan et al., 2023), and Direct Preference Optimization (DPO) (Rafailov et al., 2023). DPO, as an early initiative, leverages the mapping between the reward function and the optimal policy to bypass the need for reinforcement learning and explicit reward model learning. Still, it's equivalent to RLHF for the final solution under reverse KL regularization.

However, most existing studies focus on solutions under the constraint of reverse KL divergence, and the exploration of incorporating other divergences remains significantly lacking. To illustrate the differences among various divergence constraints, we have visualized the mode-seeking and mass-covering behaviors of reverse KL and forward KL divergences in Figure 1 The mode-seeking property of reverse KL divergence tends to reduce diversity in generation (Wiher et al., 2022; Khalifa et al., 2021; Perez et al., 2022; Glaese et al., 2022), which, although beneficial for optimizing alignment performance, may limit the model's potential (e.g., user engagement). Additionally, Santurkar et al. (2023) observe that finetuning LLMs with RLHF under the reverse KL regularization will result in a limited range of political views. As a mitigation, the inclusion of various divergences can lead to solutions with distinct characteristics (e.g., diversity), especially when the model is under-specified.

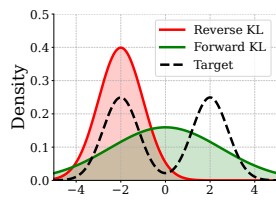

Figure 1: The mode seeking and mass covering behaviors of reverse KL and forward KL.

In this work, we generalize the DPO framework (Rafailov et al., 2023) to incorporate various divergence constraints. Different from the reverse KL divergence, we initially find that a naive derivation results in an excessively complex relationship between the reward and optimal policy for other divergences, largely due to the normalizing constant. However, by meticulously addressing the Karush–Kuhn–Tucker (KKT) conditions, specifically the complementary slackness, we demonstrate that for a class of well-known divergences, such as Jensen-Shannon divergences, forward KL divergences and $\alpha$-divergences with $\alpha \in (0, 1)$, the normalizing constant can be eliminated in the Bradley-Terry model. This results in an analytical and elegant mapping between the reward function and the optimal policy for a broad class of divergences. It further enables us to optimize the language model to align with human preferences under varying divergences constraints without needing to estimate the normalization constant. Such flexibility might allow us to explore a richer spectrum of modeling possibilities and cater to diverse application requirements

In conclusion, our key contribution is the generalization of the DPO framework to seamlessly integrate a variety of popular divergences (e.g., Jensen-Shannon divergence, forward KL divergence and $\alpha$-divergence) for regularization. Empirical results indicate that by adapting different divergence regularizations, we can achieve a nuanced balance between alignment performance (e.g., reward) and the generation diversity. This introduces greater flexibility during fine-tuning processes utilizing human preference datasets. Furthermore, comparative analyses reveal that the generalized DPO framework surpasses PPO-based methods in divergence efficiency. We also prove that the difference in the expected calibration error (ECE) is bounded by the divergence between them, emphasizing the practical benefits of improved divergence efficiency in model calibration.

## 2 RELATED WORKS

**AI Alignment** (Leike et al., 2018) is proposed as a research agenda aimed at aligning model behavior with human preferences and instruction following. Not only has AI alignment been demonstrated to be essential in ensuring safe AI behaviors, but it also improves performance on a variety of downstream tasks (Bai et al., 2022a;b; OpenAI, 2023; Touvron et al., 2023; Glaese et al., 2022). These tasks include metrics such as helpfulness (Askell et al., 2021), truthfulness (Lin et al., 2022), and non-offensiveness (Gehman et al., 2020), etc. In this context, numerous methodologies have been proposed, including red teaming (Perez et al., 2022; Korbak et al., 2023), reward modeling (Leike

et al., 2018; Gao et al., 2023), supervised fine-tuning, rejection sampling, and reinforcement learning from human/AI feedback (Ziegler et al., 2019; Ouyang et al., 2022; Bai et al., 2022b), among others. These methods largely depend on human judgment or a comprehensive set of human-written principles to provide the supervised signal. For more complex situations where humans may be incapable of evaluating, the main approach involves designing mechanisms that utilize AI to assist in evaluation by recursively decomposing the problem. This body of work includes AI debate (Irving et al., 2018), iterated amplification (Christiano et al., 2018), and recursive reward modeling (Leike et al., 2018). However, most of these methods typically involve multiple stages of training or complex interaction protocols. In contrast, we focus on proposing a single-stage algorithm that is simple to implement and computationally efficient in the setup where a human is capable of evaluating.

**Reinforcement Learning from Human Feedback (RLHF)** (Christiano et al., 2017; Bai et al., 2022a; Touvron et al., 2023; Ouyang et al., 2022) has served as a pivotal method for aligning language models, contributing significantly to the success of ChatGPT (OpenAI, 2023). Nonetheless, RLHF's complexity surpasses that of supervised learning, primarily due to the need for a distinct reward model, the quality of which decisively influences the efficacy of reinforcement fine-tuning. Any errors within this model may be exploited by the language model (Gao et al., 2023). Furthermore, RL algorithms like PPO (Schulman et al., 2017) have been proven to be less stable and more memory-intensive than supervised learning (Touvron et al., 2023). These challenges have spurred interest in alternatives to the RLHF pipeline, such as Reward Ranked FineTuning (RAFT) (Dong et al., 2023), Rank Responses to align Human Feedback (RRHF) (Yuan et al., 2023), and Direct Preference Optimization (DPO) (Rafailov et al., 2023). Despite these efforts, all the methods mentioned focus solely on solutions within the confines of reverse KL divergence regularization, leaving the potential advantages of incorporating various other divergences largely unexplored. Go et al. (2023) recently attempted to minimize the $f$-divergence for aligning language models. However, this approach necessitates the user to define the target distribution, and requires the estimation of the target distribution's normalizing constant—adding more hyperparameters and algorithmic complexity. In contrast, we propose a supervised learning method that incorporates various divergence regularizations. This method does not require the estimation of normalizing constants, the specification of the target distribution, or the use of reinforcement learning (e.g., model rollouts). Moreover, it does not involve any additional hyperparameters.

## 3 Preliminary and Backgrounds

### 3.1 Preliminary

$f$**-divergences.** For any convex function $f : \mathbb{R}^+ \to \mathbb{R}$ that satisfies $f(1) = 0$ and $f$ is strictly convex around 1, then the corresponding $f$-divergence for two distributions $p$ and $q$ is defined as

$$D_f(p, q) = \mathbb{E}_{q(x)} \left[ f \left( \frac{p(x)}{q(x)} \right) \right].$$

The $f$-divergence covers a broad class of commonly used divergences, including forward KL divergence, reverse KL divergence, Jensen-Shannon (JS) divergence and total variation distance, etc, by choosing the specific function $f$. We provide a summarization in Table 1.

**Bradley-Terry Model.** The Bradley-Terry model (Bradley & Terry, 1952) has been widely employed for pairwise comparisons. It works by assigning a real-valued "strength" parameter $p_i$ to each item $y_i$, which is then used to compute the probability that item $y_i$ outperforms item $y_j$ in a pairwise comparison by $p(y_i \succeq y_j) = p_i/(p_i + p_j)$. Yet, in practice, this model's instantiation usually adopts a specific form, denoted as

$$p(y_i \succeq y_j) = \frac{\exp(r(y_i))}{\exp(r(y_i)) + \exp(r(y_j))},$$

where $r(y_i)$ represents the "rating" or transformed "strength" of item $y_i$.

The Bradley-Terry model can be linked to Gumbel noise in the context of pairwise comparisons. The outcome depends on the comparison of $r(y_i) + \epsilon_i$ and $r(y_j) + \epsilon_j$, where $\epsilon_i$ and $\epsilon_j$ are i.i.d Gumbel-distributed noise. If $r(y_i) + \epsilon_i > r(y_j) + \epsilon_j$, $y_i$ defeats $y_j$. Noting that the difference between two Gumbel variables follows a logistic distribution, we align this with the Bradley-Terry

model's formula: $p(y_i \succeq y_j) = 1/(1 + \exp((r(y_j) - r(y_i))))$, with $p_i = \exp(r(y_i))$ and $p_j = \exp(r(y_j))$. Thus, the Bradley-Terry model represents a stochastic rank-order model influenced by Gumbel noise, which may justify the use of it for categorical distribution (e.g., language models).

**RL from Human Feedbacks.** RLHF takes place after the base model has been pretrained. It comprises three steps: 1) supervised fine-tuning, 2) reward model training, and 3) RL fine-tuning. In the RL fine-tuning process, the following objective is maximized:

$$\mathbb{E}_{x \sim \mathcal{D}, y \sim \pi_{\boldsymbol{\theta}}(\cdot|x)} [r_\varphi(y|x)] - \beta D_{\mathrm{KL}}(\pi_{\boldsymbol{\theta}}(\cdot|x)|\pi_{\mathrm{ref}}(\cdot|x)),$$

where $\mathcal{D}$ represents the dataset of prompts, $r_\varphi(\cdot|x)$ stands for the reward function learned using the Bradley-Terry model on the preference dataset, $\pi_{\mathrm{ref}}(\cdot|x)$ is the fixed reference model (typically selected to be the one post supervised fine-tuning), and $\beta$ is the coefficient of the reverse KL divergence penalty. In practice, this objective is equivalent to executing reinforcement learning under the following reward function (Ziegler et al., 2019; Bai et al., 2022a; Ouyang et al., 2022):

$$r(\cdot|x) = r_\varphi(\cdot|x) - \beta \log \left( \frac{\pi_{\boldsymbol{\theta}}(\cdot|x)}{\pi_{\mathrm{ref}}(\cdot|x)} \right).$$

**Directed Preference Optimization (DPO).** The original DPO method (Rafailov et al., 2023) establishes a functional mapping between the reward model and the optimal policy under the reverse KL divergence constraint. This allows for the direct optimization of the policy by reparameterizing the reward function using the policy (i.e., the language model) in a supervised manner,

$$r(\cdot|x) = \beta \log \frac{\pi_{\boldsymbol{\theta}}(\cdot|x)}{\pi_{\mathrm{ref}}(\cdot|x)} + \beta \log Z(x).$$

Here, $Z(x)$ is the partition function or the normalizing constant. By plugging the reward into the Bradley-Terry model, the resulting objective of DPO with reverse KL divergence is given by:

$$-\mathbb{E}_{(x,y_w,y_l) \sim \mathcal{D}} \left[ \log \sigma \left( \beta \log \frac{\pi_{\boldsymbol{\theta}}(y_w|x)}{\pi_{\mathrm{ref}}(y_w|x)} + \beta \log Z(x) - \beta \log \frac{\pi_{\boldsymbol{\theta}}(y_l|x)}{\pi_{\mathrm{ref}}(y_l|x)} - \beta \log Z(x) \right) \right], \quad (1)$$

where $\sigma$ is the sigmoid function, and the partition functions are cancelled out.

**Calibration Error.** To measure calibration, it's common to use the Expected Calibration Error (ECE) (Guo et al., 2017). Specifically, for any policy $\pi_{\boldsymbol{\theta}}(\cdot|x)$, define $\hat{P}_{\pi_{\boldsymbol{\theta}}}(x)$ as

$$\hat{P}_{\pi_{\boldsymbol{\theta}}}(x) = \pi_{\boldsymbol{\theta}}(\hat{y}|x),$$

where $\hat{y}$ is the predicted label of $x$. Given a policy $\pi_{\boldsymbol{\theta}}$, $\hat{y}$ is sampled with probability $\pi_{\boldsymbol{\theta}}(\hat{y}|x)$. Then, the ECE can be computed by,

$$\mathrm{ECE}(\boldsymbol{\theta}) = \mathbb{E}_{\hat{P}_{\pi_{\boldsymbol{\theta}}}}[|\mathbb{P}(\hat{Y} = Y|\hat{P}_{\pi_{\boldsymbol{\theta}}} = p) - p|].$$

Intuitively, if the model's confidence in its predictions closely matches the probability of its predictions being correct, then the model is well-calibrated. In the realm of LLMs, OpenAI (2023) demonstrate that RLHF degrades the model's calibration.

## 3.2 Background: Optimizing for Reverse KL Hurts Diversity

The finetuning of Large Language Models (LLMs) using Reinforcement Learning from Human Feedback (RLHF) has raised concerns regarding sample diversity. Notably, this process is prone to mode collapse (see Figure 1 for illustration). This will result in a reduction in the diversity of model outputs, as evidenced by studies from Khalifa et al. (2021), Perez et al. (2022), Go et al. (2023), and Glaese et al. (2022). One plausible explanation for mode collapse is the shift from supervised to reinforcement learning with reverse KL divergence (Song et al., 2023). Additionally, Santurkar et al. (2023) found that LLMs finetuned with RLHF under reverse KL divergence regularization can express a limited range of political views. Given these observations, there's a clear need to investigate alternative divergence regularization methods to maintain the diversity and integrity of LLM outputs, and understand the tradeoff.

Table 1: Summary of some commonly used $f$-divergences including their derivatives.

| $f$-divergence | $f(u)$ | $f'(u)$ | $0 \notin$ Domain of $f'(u)$ |
|---|---|---|---|
| $\alpha$-divergence ($\alpha \in (0,1)$) | $(u^{1-\alpha} - (1-\alpha)u - \alpha)/(\alpha(\alpha-1))$ | $(1 - u^{-\alpha})/\alpha$ | ✓ |
| Reverse KL ($\alpha = 0$) | $u \log u$ | $\log u + 1$ | ✓ |
| Forward KL ($\alpha = 1$) | $-\log u$ | $-1/u$ | ✓ |
| JS-divergence | $u \log u - (u+1)\log((u+1)/2)$ | $\log(2u/(1+u))$ | ✓ |
| Total Variation | $\frac{1}{2}|u-1|$ | $u > 1 ? \frac{1}{2} : -\frac{1}{2}$ | ✗ |
| Chi-squared | $(u-1)^2$ | $2(u-1)$ | ✗ |

## 4 METHOD: DIRECT PREFERENCE OPTIMIZATION UNDER $f$-DIVERGENCE

During the RL finetuning process, it is common to regularize the finetuned model to stay "close" to the original model (or reference model) measured by the KL divergence. Typically, reverse KL divergence (i.e., $D_{\text{KL}}(\pi_{\boldsymbol{\theta}} \| \pi_{\text{ref}})$) is a default choice. However, the mode-seeking behavior will lead to low diversity in the generations. Therefore, to balance the alignment performance (e.g., accuracy or reward) and diversity, we consider a more broad class of divergence regularization, namely, the $f$-divergence, which covers many commonly used divergences, such as forward KL, reverse KL, JS divergence and $\alpha$-divergence, etc.

When given the reward function $r(y|x)$, the base model is fine-tuned using reinforcement learning to maximize the reward under certain constraints. From an optimization perspective, this step is equivalent to solving the following constrained optimization problem:

$$\max_{\pi} \mathbb{E}_{\pi}[r(y|x)] - \beta D_f(\pi, \pi_{\text{ref}}) \quad s.t. \quad \sum_y \pi(y|x) = 1 \text{ and } \pi(y|x) \geq 0 \, \forall y.$$

The two constraints are introduced to ensure that the solution is a valid distribution, though in practice we don't need to explicitly deal with them. To solve the constrained problem, we can apply the Lagrange multiplier, which gives us

$$\mathcal{L}(\pi, \lambda, \alpha) = \mathbb{E}_{\pi}[r(y|x)] - \beta \mathbb{E}_{\pi_{\text{ref}}}\left[f\left(\frac{\pi(y|x)}{\pi_{\text{ref}}(y|x)}\right)\right] - \lambda\left(\sum_y \pi(y|x) - 1\right) + \sum_y \alpha(y)\pi(y|x),$$

where $\lambda$ and $\alpha(y)$ are the dual variables. For such problems, we can derive the closed-form solution for $\pi^{\star}$, which optimally solves the above problem:

$$\pi^{\star}(y|x) = \frac{1}{Z(x)}\pi_{\text{ref}}(y|x)(f')^{-1}\left(\frac{r(y|x)}{\beta}\right),$$

where $Z(x)$ is the normalization constant, and $(f')^{-1}$ is the inverse function of $f'$. By solving the equation for $r(y|x)$, we establish the following relationship between $r(y|x)$ and $\pi^{\star}(y|x)$,

$$r(y|x) = \beta f'\left(\frac{\pi^{\star}(y|x)}{\pi_{\text{ref}}(y|x)} \cdot Z(x)\right).$$

When $D_f$ is the reverse KL divergence, we have $f'(u) = \log u + 1$ from Table 1. Thus, $r(y|x) = \beta \log(\pi^{\star}(y|x)/\pi_{\text{ref}}(y|x)) + \log Z(x) + 1$. Since the BT model is defined by $p(y_w \succ y_l) = \sigma(r(y_w|x) - r(y_l|x))$, the $\log Z(x)$, which appears as an additive term in the reward, will be canceled out for the reverse KL divergence (see Equation 1), but this is not the case for other divergences. This situation is discouraging, as estimating the $Z(x)$ requires multiple samples, which may be computationally expensive and may exhibit high variance if not handled properly.

Fortunately, we will show that, by carefully analyzing the normalization constant $Z(x)$, we can derive a closed-form solution for many other (but not all) divergences as well, without the need to estimate the normalization constant $Z(x)$. To achieve this, we can first rewrite $\pi^{\star}(y|x)$ in the following format using the dual variables $\lambda$ and $\alpha(y)$:

$$\pi^{\star}(y|x) = \pi_{\text{ref}}(y|x)(f')^{-1}\left(\frac{r(y|x) - \lambda + \alpha(y)}{\beta}\right).$$

---

**Algorithm 1** Direct Preference Optimization with $f$-divergences (DPO-$f$)

---

**Require:** Preference dataset $\mathcal{D}$, batch size $b$, constraint coefficient $\beta$, divergence function $f$, and learning rate $\eta$.

1: Initialize model $\pi_{\boldsymbol{\theta}_0}$ with supervised finetuning on $\mathcal{D}$.
2: **for** $n = 1 \dots N$ iterations **do**
3:     Sample a batch $\mathcal{B} = \{(x_i, y_i^w, y_i^l)\}_{i=1}^b$ from $\mathcal{D}$.
4:     Compute the loss using the equation 3 with the chosen function $f$.
5:     Compute the gradient and update the model $\boldsymbol{\theta}_t \leftarrow \boldsymbol{\theta}_{t-1} - \eta \nabla_{\boldsymbol{\theta}} \mathcal{L}(\boldsymbol{\theta}_{t-1}, \mathcal{B})$.
6: **end for**
7: **return** Final model $\pi_{\boldsymbol{\theta}}$.

---

However, the term $\alpha(y)$ depends on $y$, and cannot be canceled out, making it hard to compute. Next, we will show that for a class of $f$-divergences, we must have $\alpha(y) = 0$, and thus we can represent the reward using only the trainable policy, the reference policy, and a constant that is independent of $y$. This result is summarized in the following theorem.

**Theorem 1.** *If $\pi_{\mathrm{ref}}(y|x) > 0$ for any valid $x$ and $f'$ is invertible with $0 \notin dom(f')$, the reward class that is consistent with the Bradley-Terry model can be reparameterized using the policy model $\pi(y|x)$ and a reference model $\pi_{\mathrm{ref}}(y|x)$ as*

$$r(y|x) = \beta f'\left(\frac{\pi^\star(y|x)}{\pi_{\mathrm{ref}}(y|x)}\right) + \mathrm{const.} \tag{2}$$

Theorem 1 holds mainly due to the complementary slackness in the KKT conditions, and the proof can be found in Appendix F. Moreover, the requirement $0 \notin \mathrm{dom}(f)$ already covers many commonly used divergences, including forward KL divergence, Jensen-Shannon divergence, reverse KL divergence, and $\alpha$-divergence with $0 < \alpha < 1$. Lastly, the constant is independent of $y$ and appears as an additive term in the reward function, and thus it will be cancelled out when plugged into the Bradley-Terry model.

Now, for a pair of examples $(x, y_w)$ and $(x, y_l)$, we can plug the reward from Equation 2 into the Bradley-Terry model, which gives us the following expression,

$$p(y_w \succeq y_l|x) = \sigma\left(\beta f'\left(\frac{\pi^\star(y_w|x)}{\pi_{\mathrm{ref}}(y_w|x)}\right) - \beta f'\left(\frac{\pi^\star(y_l|x)}{\pi_{\mathrm{ref}}(y_l|x)}\right)\right).$$

Hence, for a preference dataset $\mathcal{D}$, we train the model $\pi_{\boldsymbol{\theta}}$ (replacing $\pi^\star$ in the above equation) by minimizing the following negative log-likelihood loss,

$$\mathcal{L}(\boldsymbol{\theta}, \mathcal{D}) = \mathbb{E}_{(x, y_w, y_l) \sim \mathcal{D}}\left[-\log \sigma\left(\beta f'\left(\frac{\pi_{\boldsymbol{\theta}}(y_w|x)}{\pi_{\mathrm{ref}}(y_w|x)}\right) - \beta f'\left(\frac{\pi_{\boldsymbol{\theta}}(y_l|x)}{\pi_{\mathrm{ref}}(y_l|x)}\right)\right)\right]. \tag{3}$$

The above helps us to solve the RL finetuning problem via a supervised learning approach under a broad class of divergence constraints, which is more stable and efficient to optimize in contrast to the reinforcement learning counter-part. Our results generalize the DPO (Rafailov et al., 2023) to a more broad class of divergence regularization. The full algorithm is summarized in Algorithm 1.

## 5 EXPERIMENTS

### 5.1 EXPERIMENTAL SETUP

**Baselines and Datasets.** For the experiments, we adopt three datasets, including IMDB-sentiment dataset (Maas et al., 2011), Anthropic HH dataset (Bai et al., 2022a) and MT-bench (Zheng et al., 2023) for evaluation. Our primary baseline approach is PPO with different $f$-divergences. Upon experimentation (see Appendix A), it was observed that incorporating a divergence penalty in the reward for variants of PPO, such as those with forward KL divergence and JS divergence, induces training instability. The reason being that the value ranges of these divergences are considerably larger than those of reverse KL divergence. This discrepancy causes substantial challenges when trying to learn a accurate value function in PPO. To address the issue, we further propose a modified

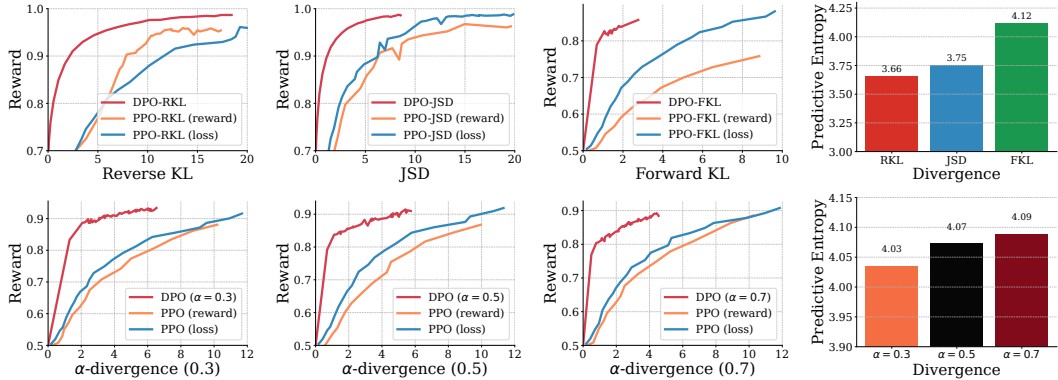

Figure 2: Comparisons between DPO with various $f$-divergences and PPO in terms of the frontier of divergence vs reward. To be noted, 'reward' means we add the divergence penalty in the reward, and 'loss' means we add the divergence penalty in the loss.

variant of PPO as an additional baseline. Instead of integrating the divergence penalty into the reward function, we treat it as a regularization term separately. The new objective (i.e., PPO (loss)) is optimized separately by PPO and SGD,

$$\text{PPO (reward):} \qquad \underbrace{\mathbb{E}_{x\sim\mathcal{D},y\sim\pi_{\boldsymbol{\theta}}(\cdot|x)}\left[r_{\varphi}(y|x) - \beta f(\pi_{\boldsymbol{\theta}}(y|x)/\pi_{\text{ref}}(y|x))\right]}_{\text{Optimized by PPO}},$$

$$\text{PPO (loss):} \qquad \underbrace{\mathbb{E}_{x\sim\mathcal{D},y\sim\pi_{\boldsymbol{\theta}}(\cdot|x)}\left[r_{\varphi}(y|x)\right]}_{\text{Optimized by PPO}} - \underbrace{\beta D_f(\pi_{\text{ref}}(\cdot|x), \pi_{\boldsymbol{\theta}}(\cdot|x))}_{\text{Optimized by SGD}}.$$

For the sake of differentiation, we term the conventional PPO method as PPO (reward) – indicating the inclusion of divergence penalty in the reward. In contrast, the variant where the divergence regularization is treated separately is denoted as PPO (loss).

## 5.2 Experiments on IMDB Dataset

Our initial experiments were performed on the IMDB-sentiment dataset (Maas et al., 2011) for comparing $f$-DPO against PPO. Following the setup in trlx (CarperAI, 2023), we utilized GPT-2-large (Radford et al., 2019) as our base model for fine-tuning, and the SiEBERT model (Hartmann et al., 2023)—a fine-tuned model of RoBERTa-large (Liu et al., 2019)—was employed for reward computation. For PPO, we explored the divergence coefficient in $\{0.01, 0.03, 0.1, 0.3\}$ for both PPO variants, each using ground-truth rewards. Our PPO implementation is based on the trlx library. Additionally, we adapted the official implementation of DPO with $f$-divergences from Rafailov et al. (2023), setting $\beta$ at 0.1. Please note that PPO utilizes the ground-truth reward during training.

The results are depicted in Figure 2, and we also visualize the trade-off curve in Figure 3. Our observations indicate that DPO with $f$-divergences outclasses both PPO implementations in terms of divergence versus reward on the frontier, thus establishing greater divergence-efficiency. Additionally, PPO with a divergence penalty in loss (i.e., PPO (loss)) surpasses PPO with a divergence penalty in reward for both JSD, forward KL and $\alpha$-divergences. This discrepancy arises due to the JSD divergence penalty and forward KL penalty fluctuating more significantly than the reverse KL penalty, which consequently introduces instability in learning the value function in PPO. Further experimental results can be found in the

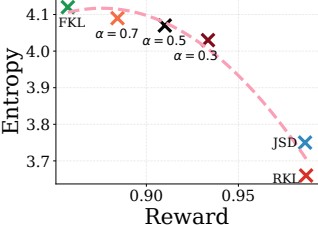

Figure 3: The reward and entropy tradeoff of $f$-DPO for different divergences.

Appendix A. Finally, we note that reverse KL achieves the lowest predictive entropy due to its mode-seeking property, while Forward KL exhibits the highest predictive entropy. JSD maintains a balance between the two. $\alpha$-divergence interpolates between the JSD and forward KL divergence. This observation aligns with the property of these divergence regularization as well as those in Go et al. (2023), although our framework does not necessitates manual specification of the target distribution or estimation of the normalizing constant under various divergence regularizations.

Table 2: Comparison of JSD, RKL, FKL and some $\alpha$-divergences in terms of Alignment and Diversity on Anthropic HH. The $\uparrow$ indicates that higher is better, and $\downarrow$ means lower is better.

| Divergences | Alignment | Diversity | | | |
| --- | --- | --- | --- | --- | --- |
| | Accuracy (%) $\uparrow$ | Entropy $\uparrow$ | Self-Bleu $\downarrow$ | Distinct-1 $\uparrow$ | Distinct-2 $\uparrow$ |
| RKL | **67.19** | 12.25 | 0.880 | 0.021 | 0.151 |
| JSD | 66.80 | 12.31 | 0.878 | 0.021 | 0.159 |
| $\alpha = 0.3$ | 59.77 | 12.85 | 0.849 | 0.026 | 0.199 |
| $\alpha = 0.5$ | 61.72 | 12.90 | 0.841 | 0.028 | 0.206 |
| $\alpha = 0.7$ | 57.42 | 12.98 | 0.839 | 0.027 | 0.202 |
| FKL | 54.30 | **13.01** | **0.834** | **0.029** | **0.210** |

### 5.3 EXPERIMENTS ON ANTHROPIC HH DATASET AND MT-BENCH

Our next set of experiments was conducted on the Anthropic HH dataset (Bai et al., 2022a). We adopted the Pythia 2.8B model from Biderman et al. (2023) as our base model. The training configuration follows from Rafailov et al. (2023)[1]. The goals of these experiments were to study: 1) how different divergence regularizations impact the trade-off between alignment and diversity in the generated responses, and 2) how $f$-DPO compares to its PPO counterparts. For the first part of the experiments, we utilized automatic metrics for evaluation, while for the second part, we relied on the GPT-4 evaluation. For PPO, we adopt the PPO (loss).

The results, in terms of alignment accuracy and diversity, are presented in Table 2. To measure diversity, we generated 25 responses using nucleus sampling (Holtzman et al., 2020) with $p = 0.95$ for each prompt in the test set of the Anthropic HH dataset using temperatures of $0.6, 1.0, 1.4$, following Touvron et al. (2023). The results for temperatures $0.6$ and $1.4$ can be found in Appendix E.1. To compute metrics, we employed the predictive entropy, self-bleu (Zhu et al., 2018) and distinct-n (Li et al., 2016). Consistent with the findings in section 5.2, we observed that reverse KL divergence achieves the highest accuracy but the lowest diversity in generation. Adjusting the divergence regularization allows us to trade-off between alignment accuracy and diversity.

By tailoring outputs to specific application needs, it offers a customizable balance between accuracy and diversity. This might not only enhance robustness against unfamiliar inputs but also boost user engagement by preventing monotonous interactions.

To compare $f$-DPO with PPO in terms of generation quality, we conducted a pairwise comparison using MT-Bench (Zheng et al., 2023). MT-Bench is a GPT-4-based evaluation benchmark for LLMs that achieves over $80\%$ agreement with human preference judgments on LLM generation quality. MT-Bench includes a series of open-ended questions that assess LLM capabilities in multi-turn conversations and instruction-following, which are often considered important factors for human preference. In accordance with the official MT-Bench implementation (Zheng et al., 2023),[2] we sampled responses with a temperature setting of $0.7$ and limited the maximum number of newly generated tokens to 1024. For additional details about MT-bench, we refer readers to the original paper. The GPT-4 evaluation results are provided in Figure 4. From these results, it is evident that extending DPO with $f$-divergence achieves performance that is comparable to, and in some cases significantly better than, that of PPO. We've also provided the comparisons between DPO under different $f$-divergence in Appendix E.2.

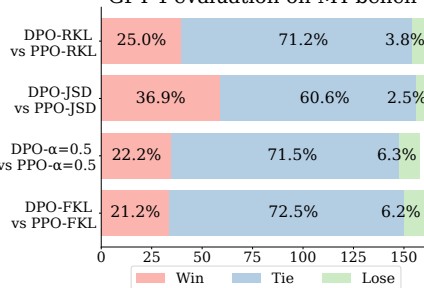

Figure 4: MT-Bench comparison between $f$-DPO and PPO under different divergences. The win, tie and lose rates are evaluated based on GPT-4.

### 5.4 EXPERIMENTS ON CALIBRATION

In our last set of experiments, we sought to explore the advantages of divergence efficiency. Our observations revealed that $f$-DPO achieves a smaller divergence than PPO while attaining comparable performance. Previous studies, as referenced by OpenAI (2023), suggest that RLHF adversely

---

[1] https://github.com/eric-mitchell/direct-preference-optimization
[2] https://github.com/lm-sys/FastChat/blob/main/fastchat/llm_judge/gen_model_answer.py

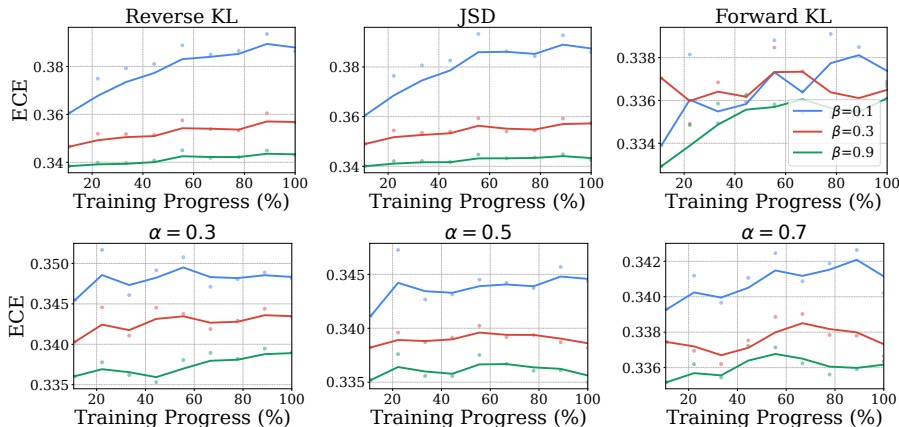

Figure 5: Evolution of Expected Calibration Error (ECE) across training steps for three different divergence regularizations: Reverse KL, JSD, Forward KL and $\alpha$-divergence ($\alpha = 0.3, 0.5$ and $0.7$). Each subplot represents the ECE values for varying regularization parameters $\beta = 0.1, 0.3$ and $0.9$ with exponential smoothing.

affects the calibration performance of GPT-4. This prompts the question: is there a correlation between the divergence of the base model and that of the finetuned version, and the calibration error? To elucidate this, we presented the following theorem,

**Theorem 2.** *Suppose $\pi_{\boldsymbol{\theta}_1}(\cdot|x)$ and $\pi_{\boldsymbol{\theta}_2}(\cdot|x)$ be two policies. Let $D_f$ be any $f$-divergence such that $f$ is strictly convex.*

$$\text{ECE}(\boldsymbol{\theta}_1) - \text{ECE}(\boldsymbol{\theta}_2) \leq 2\mathbb{E}_X[\psi_f(D_f(\pi_{\boldsymbol{\theta}_1}(\cdot|x), \pi_{\boldsymbol{\theta}_2}(\cdot|x)))]$$

*where $\psi_f$ is a real-valued function such that $\lim_{x\downarrow 0} \psi_f(x) = 0$.*

**Remark 1.** *For JSD, we have $\text{ECE}(\boldsymbol{\theta}_1) - \text{ECE}(\boldsymbol{\theta}_2) \leq \mathbb{E}_X\left[4\sqrt{2D_{JS}(\pi_{\boldsymbol{\theta}_1}(\cdot|x), \pi_{\boldsymbol{\theta}_2}(\cdot|x))}\right]$.*

**Remark 2.** *For KL, we have $\text{ECE}(\boldsymbol{\theta}_1) - \text{ECE}(\boldsymbol{\theta}_2) \leq \mathbb{E}_X\left[2\sqrt{2D_{KL}(\pi_{\boldsymbol{\theta}_1}(\cdot|x), \pi_{\boldsymbol{\theta}_2}(\cdot|x))}\right]$.*

Theorem 2, detailed further in Appendix G, establishes a relationship between Expected Calibration Error (ECE) difference and $f$-divergences. Specifically, the difference in ECE between two models can be bounded by the $f$-divergence. Thus, if the base model exhibits good calibration (i.e., small ECE), a smaller $f$-divergence suggests that the finetuned model is similarly well-calibrated.

To validate the theoretical findings, we conducted experiments to explore the impact of various $f$-divergences on calibration errors. We evaluated calibration error on Anthropic HH dataset (Bai et al., 2022a) with the Pythia 2.8B model from Biderman et al. (2023) as our base model. The model was finetuned using $f$-DPO in the same way as in Section 5.3. We treat the task as a binary prediction problem. For predictive probabilities, we use the exponentials of normalized scores where the scores are computed for chosen and rejected strings conditioned on input prompts. The results are shown in Figure 5. It is apparent that increased regularization parameters can restrict the extent to which the calibration error can increase. On the other hand, as training progresses, the calibration error increases as well. Similar trends on calibration have been discovered in OpenAI (2023) as well.

## 6 CONCLUSION

In this study, we introduced a generalized framework for DPO, elegantly incorporating a spectrum of divergence constraints. Our empirical results show that while the reverse KL divergence typically offers superior alignment performance, it compromises generation diversity. By adjusting the divergence regularization, we can achieve a nuanced balance between alignment and generation diversity. Notably, the $f$-DPO framework demonstrates greater divergence efficiency than traditional PPO methods. We further established that the difference in the expected calibration error of two models can be bounded by their divergence, underscoring the advantages of enhanced divergence efficiency. For practitioners, we recommend using JS divergence as the first choice, as it generally generates more diversified responses and is more favored by GPT-4 than reverse KL divergence. Looking forward, we aim to explore the integration of other divergences not well addressed by our present formulation, such as the total variation distance.

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

## A  THE CAUSE OF INSTABILITY IN PPO WITH DIVERGENCE PENALTY IN REWARDS

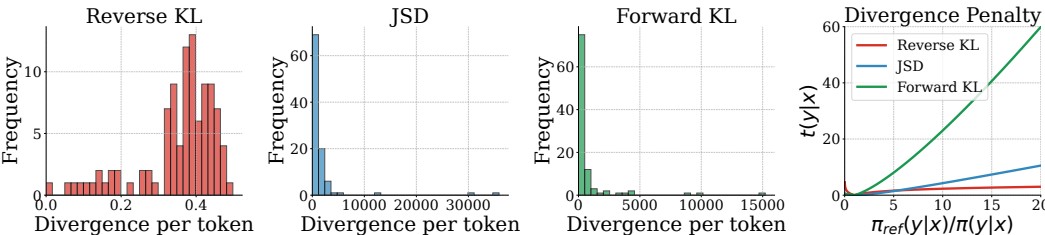

Figure 6: Visualization of the divergence penalty for reverse KL, JDS and forward KL.

In RLHF, it is a common practice to impose the KL-divergence as part of the reward function. However, this practice can lead to instability in the optimization process for certain types of divergence measures, such as the forward KL divergence, Jensen-Shannon (JS) divergence, etc. To illustrate this, consider the following reward functions:

$$r_{\text{RKL}}(y|x) = r(y|x) + \beta \cdot \log t(y|x), \tag{4}$$

$$r_{\text{JS}}(y|x) = r(y|x) - \beta \cdot \left( t(y|x) \log t(y|x) - (t(y|x) + 1) \log \left( \frac{t(y|x) + 1}{2} \right) \right), \tag{5}$$

$$r_{\text{FKL}}(y|x) = r(y|x) - \beta \cdot t(y|x) \log t(y|x), \tag{6}$$

where $t(y|x) = \pi_{\text{ref}}(y|x)/\pi(y|x)$. Figure 6 illustrates the curves of the above three divergence penalties as well as the divergence per token computed on the the IMDB dataset during training using PPO with the divergence penalty included in the reward. We oberve that the forward KL divergence penalty will grow much faster than the other two, and the reverse KL divergence penalty grows the slowest. This difference make the reverse KL divergence more numerically stable then the other two and thus makes the learning of the value function much easier.

## B  IMPLEMENTING PPO UNDER $f$-DIVERGENCE CONSTRAINT

To implement PPO under various $f$-divergence, we need to estimate the $f$-divergence using samples. Therefore, an unbiased estimator with low variance is desired. Following Schulman (2020), by denoting $r(x) = p(x)/q(x)$ to be the ratio between two distributions $p$ and $q$, then the following estimator for the $f$-divergence between $p$ and $q$ is unbiased and has low variance,

$$\mathbb{E}_X \left[ f(r(x)) - f'(1)(r(x) - 1) \right].$$

Therefore, we adopt the above estimator for the $f$-divergence when we implement PPO.

## C  ADDITIONAL DISCUSSIONS ON THE CHOICE OF DIVERGENCES

Different divergences may impose different properties in the optimized solution as we demonstrated through experiments. In pratice, if there is only one model allowed, we would suggest initially considering the Jensen-Shannon (JS) divergence. Our recommendation is based on its ability to yield more diverse responses compared to the commonly used reverse KL divergence. This preference is supported by our findings, where JS divergence can result in a more diversified response as measured by different diversity metrics. Additionally, despite underperforming in reverse KL in reward metrics, it was favored in the GPT-4 evaluation, as illustrated in Figure 7 of the appendix. This observation also aligns with recent studies, such as Sun et al. (2023).

To accommodate diverse application needs, another strategy could involve training multiple models, each employing a different divergence, such as JS divergence, $\alpha$ divergence (with $\alpha$ values ranging from 0.1 to 0.9), and forward KL divergence. This approach allows users to experiment and choose the divergence that best aligns with their specific requirements or preferences.

## D  ADDITIONAL DISCUSSIONS ON RKL VS FKL

The distinction between Reverse KL Divergence (RKL) and Forward KL Divergence (FKL) primarily lies in their inherent properties: RKL is mode-seeking, while FKL is mass-covering as we shown in Figure 1. This can be discerned from their mathematical formulations. RKL is defined as $E_q[\log(q(x)/p(x))]$, where $q(x)$ is the distribution being optimized, typically representing the language model undergoing fine-tuning. In the RKL scenario, $q(x)$ may assign zero probability to values where $p(x) > 0$. In contrast, FKL is defined as $E_p[\log(p(x)/q(x))]$, which necessitates that $q(x)$ assigns a non-zero probability to all values where $p(x) > 0$. This requirement inherently encourages FKL to promote a distribution $q(x)$ that covers the entire range of $p(x)$, thereby enhancing diversity.

## E  ADDITIONAL EXPERIMENTAL RESULTS

### E.1  GENERATION DIVERSITY ON ANTHROPIC HH WITH DIFFERENT TEMPERATURES

Table 3: Comparison of divergences on Anthropic HH with temperature= 0.6.

| Divergences | Self-Bleu ↓ | Distinct-1 ↑ | Distinct-2 ↑ |
|---|---|---|---|
| RKL | 0.8667 | 0.0092 | 0.0615 |
| JSD | 0.8679 | 0.0099 | 0.0662 |
| $\alpha = 0.3$ | 0.8611 | 0.0136 | 0.0899 |
| $\alpha = 0.5$ | 0.8579 | **0.0148** | **0.0950** |
| $\alpha = 0.7$ | 0.8563 | 0.0139 | 0.0905 |
| FKL | **0.8515** | 0.0142 | 0.0926 |

Table 4: Comparison of divergences on Anthropic HH with temperature= 1.4.

| Divergences | Self-Bleu ↓ | Distinct-1 ↑ | Distinct-2 ↑ |
|---|---|---|---|
| RKL | 0.7975 | 0.0973 | 0.6574 |
| JSD | 0.7995 | 0.1025 | 0.6439 |
| $\alpha = 0.3$ | 0.7759 | 0.1107 | 0.6952 |
| $\alpha = 0.5$ | 0.7603 | 0.1101 | 0.6692 |
| $\alpha = 0.7$ | **0.7537** | 0.1151 | 0.6659 |
| FKL | 0.7566 | **0.1233** | **0.7082** |

In this section, we report the additional results on evaluating the generation diversity of models trained with different divergences. The temperature were set to be 0.6 and 1.4 following Touvron et al. (2023). For each prompt, we sampled 25 responses. The results can be found in Tables 3 and 4. We found that the pattern is similar to the one observed in the main paper, where we set the temperature to be 1.0.

### E.2  GPT-4 EVALUATIONS ON MT-BENCH FOR DPO WITH DIFFERENT DIVERGENCES

We further provide a comparison between DPO using different divergences, with GPT-4 serving as the referee on MT-bench. We observe that while DPO with reverse KL outperforms DPO with JSD in terms of alignment accuracy on Anthropic HH, it underperforms when evaluated by GPT-4. Additionally, DPO with JSD performs better than DPO with $\alpha$-divergence. Lastly, DPO with $\alpha$-divergence performs slightly better than DPO with forward KL divergence. These results largely align with our expectations.

## F  PROOFS

**Theorem 1.** *If $\pi_{\text{ref}}(y|x) > 0$ for any valid $x$ and $f'$ is invertible with $0 \notin dom(f')$, the reward class that is consistent with the Bradley-Terry model can be reparameterized using the policy model*

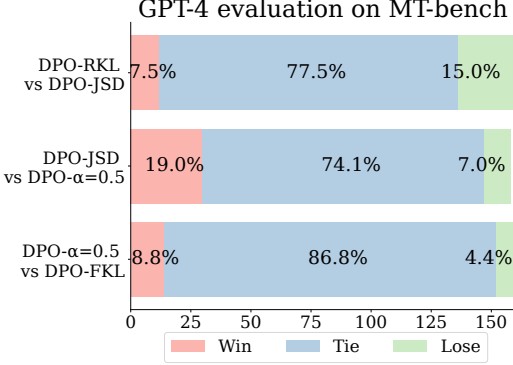

Figure 7: Comparing DPO with different divergence regularizations using GPT-4 on MT-Bench.

$\pi(y|x)$ *and a reference model* $\pi_{\text{ref}}(y|x)$ *as*

$$r(y|x) = \beta f' \left( \frac{\pi^\star(y|x)}{\pi_{\text{ref}}(y|x)} \right) + \text{const.} \tag{2}$$

*Proof.* The Karush-Kuhn-Tucker (KKT) conditions for the given optimization problem can be stated as follows:

1. **Stationarity Condition**:

   This condition requires that the gradient of the Lagrangian with respect to the primal variables be zero:
   $$\nabla_{\pi(y|x)} \mathcal{L}(\pi, \lambda, \alpha) = 0, \ \forall y.$$

   By setting the derivative of the Lagrangian with respect to $\pi(y|x)$ to zero, we obtain:
   $$r(y|x) - \beta \frac{\partial}{\partial \pi(y|x)} \mathbb{E}_{\pi_{\text{ref}}} \left[ f \left( \frac{\pi(y|x)}{\pi_{\text{ref}}(y|x)} \right) \right] - \lambda + \alpha(y) = 0, \ \forall y.$$

2. **Primal Feasibility**:

   This condition requires that the solution satisfy the original constraints of the problem:
   $$\sum_y \pi(y|x) = 1, \quad \text{and} \quad \pi(y|x) \geq 0 \ \forall y.$$

3. **Dual Feasibility**:

   This condition requires that the Lagrange multipliers corresponding to inequality constraints are non-negative:
   $$\alpha(y) \geq 0, \ \forall y.$$

4. **Complementary Slackness**:

   This condition requires that for each inequality constraint, either the constraint is satisfied with equality, or the corresponding Lagrange multiplier is zero:
   $$\alpha(y)\pi(y|x) = 0, \ \forall y.$$

In this context, $\pi(y|x)$ is the primal variable we're optimizing, $\lambda$ is the Lagrange multiplier for the equality constraint, and $\alpha(y)$ are the Lagrange multipliers for the inequality constraints.

To derive the final solution to the given problem, we first use the stationarity condition to obtain an equation that relates $\pi(y|x)$, $r(y|x)$, $f$, $\lambda$, and $\alpha(y)$. This will involve taking the derivative of $f \left( \frac{\pi(y|x)}{\pi_{\text{ref}}(y|x)} \right)$ with respect to $\pi(y|x)$, which will depend on the specific form of the function $f$.

We denote the derivative of $f$ with respect to its argument as $f'$, and that the derivative of $r(y|x)$ with respect to $\pi(y|x)$ is zero (since $r(y|x)$ does not explicitly depend on $\pi(y|x)$), we get:

$$r(y|x) - \beta f'\left(\frac{\pi(y|x)}{\pi_{\text{ref}}(y|x)}\right) - \lambda + \alpha(y) = 0.$$

We can solve this for $\pi(y|x)$, assuming that the inverse of $f'$ exists:

$$\pi(y|x) = \pi_{\text{ref}}(y|x)f'^{-1}\left(\frac{r(y|x) - \lambda + \alpha(y)}{\beta}\right).$$

However, we have to note that this is under the assumptions that $f'$ is invertible, and that the inverse maps the argument into a domain where the original function $f$ is defined and differentiable.

Next, we can use the primal feasibility condition to get an equation that will help determine the values of $\lambda$ and $\alpha(y)$. Substituting the expression for $\pi(y|x)$ into the constraint $\sum_y \pi(y|x) = 1$, we obtain an equation that can be solved for $\lambda$:

$$\sum_y \pi_{\text{ref}}(y|x)f'^{-1}\left(\frac{r(y|x) - \lambda + \alpha(y)}{\beta}\right) = 1.$$

This equation is likely to be nonlinear and might need numerical methods to solve for $\lambda$ and $\alpha(y)$. Also, we need to ensure $\pi(y|x) \geq 0$ for all $y$, which could place further restrictions on the possible values of $\lambda$ and $\alpha(y)$.

Finally, the complementary slackness condition $\alpha(y)\pi(y|x) = 0$ for all $y$ will eliminate some solutions, because for each $y$, either $\alpha(y) = 0$ or $\pi(y|x) = 0$ must hold.

Therefore, we can write out the reward function as a function of the policy, i.e.,

$$r(y|x) = \beta f'\left(\frac{\pi(y|x)}{\pi_{\text{ref}}(y|x)}\right) + \lambda - \alpha(y).$$

The complementary slackness requires that

$$\pi(y|x)\alpha(y) = 0 \ \ \forall y.$$

Hence, for those function $f$ that $0 \notin \text{dom}(f')$ with the assumption that $\pi_{\text{ref}}(y|x) > 0$ almost surely, we must have $\alpha(y) = 0 \ \forall y$. In particular, the reverse KL, forward KL and JS divergences are among this category, See Table 1. Thus, we can simplify the reward function as,

$$r(y|x) = \beta f'\left(\frac{\pi(y|x)}{\pi_{\text{ref}}(y|x)}\right) + \lambda,$$

where $\lambda$ depends only on $x$ and thus can be treated as an additive constant, which will be cancelled out in the BT model.

$\square$

To give some examples, for reverse KL divergence, we have

$$r(y|x) = \beta \log \frac{\pi(y|x)}{\pi_{\text{ref}}(y|x)} + \text{const.}$$

For forward KL divergence, we have

$$r(y|x) = -\beta \frac{\pi_{\text{ref}}(y|x)}{\pi(y|x)} + \text{const.}$$

For JS divergence, we have

$$r(y|x) = \beta \log \frac{2\pi(y|x)}{\pi_{\text{ref}}(y|x) + \pi(y|x)} + \text{const.}$$

## G  ON CALIBRATION AND $f$-DIVERGENCE

To measure calibration, we adopt the definition of Expected Calibration Error (ECE) (Guo et al., 2017). Specifically, for any policy $\pi_{\boldsymbol{\theta}}(\cdot|x)$, define $\hat{P}_{\pi_{\boldsymbol{\theta}}}(x)$ as

$$\hat{P}_{\pi_{\boldsymbol{\theta}}}(x) = \pi_{\boldsymbol{\theta}}(\hat{y}|x)$$

where $\hat{y}$ is the predicted label of $x$. Given a policy $\pi_{\boldsymbol{\theta}}$, $\hat{y}$ is sampled with probability $\pi_{\boldsymbol{\theta}}(\hat{y}|x)$.

Then ECE can be defined as follows:

$$\text{ECE}(\boldsymbol{\theta}) = \mathbb{E}_{\hat{P}_{\pi_{\boldsymbol{\theta}}}}[|\mathbb{P}(\hat{Y} = Y|\hat{P}_{\pi_{\boldsymbol{\theta}}} = p) - p|]$$

**Remark 3.** *Here, we consider our policy to be stochastic. This is different from the definition in (Guo et al., 2017) where $\hat{y}$ is chosen to be the one with the highest probabilities.*

The following theorem bound the differences of ECE by $f$ divergences of policies.

**Theorem 2.** *Suppose $\pi_{\boldsymbol{\theta}_1}(\cdot|x)$ and $\pi_{\boldsymbol{\theta}_2}(\cdot|x)$ be two policies. Let $D_f$ be any $f$-divergence such that $f$ is strictly convex.*

$$\text{ECE}(\boldsymbol{\theta}_1) - \text{ECE}(\boldsymbol{\theta}_2) \leq 2\mathbb{E}_X[\psi_f(D_f(\pi_{\boldsymbol{\theta}_1}(\cdot|x), \pi_{\boldsymbol{\theta}_2}(\cdot|x)))]$$

*where $\psi_f$ is a real-valued function such that $\lim_{x\downarrow 0} \psi_f(x) = 0$.*

*Proof.* By the tower rule, we know that

$$\text{ECE}(\boldsymbol{\theta}) = \mathbb{E}_X\left[\mathbb{E}_{\hat{P}_{\pi_{\boldsymbol{\theta}}}|X}\left[\left|\mathbb{P}(\hat{Y} = Y|\hat{P}_{\pi_{\boldsymbol{\theta}}} = p, X = x) - p\right|\right]\right]$$

$$= \mathbb{E}_X\left[\mathbb{E}_{\hat{P}_{\pi_{\boldsymbol{\theta}}}(X)}\left[\left|\mathbb{P}(\hat{Y} = Y|\hat{P}_{\pi_{\boldsymbol{\theta}}} = p, X = x) - p\right|\right]\right]$$

$$= \mathbb{E}_X\left[\sum_{\hat{y}} \pi_{\boldsymbol{\theta}}(\hat{y}|x)\left|\mathbb{P}\left(Y = \hat{y}\middle|\hat{P}_{\pi_{\boldsymbol{\theta}}} = \pi_{\boldsymbol{\theta}}(\hat{y}|x), X = x\right) - \pi_{\boldsymbol{\theta}}(\hat{y}|x)\right|\right]$$

$$= \mathbb{E}_X\left[\sum_{\hat{y}} \pi_{\boldsymbol{\theta}}(\hat{y}|x)\left|\mathbb{P}(Y = \hat{y}|X = x) - \pi_{\boldsymbol{\theta}}(\hat{y}|x)\right|\right]$$

Let $\pi(y|x)$ be the conditional distribution of ground truth. Then,

$$\text{ECE}(\boldsymbol{\theta}) = \mathbb{E}_X\left[\langle|\pi(\cdot|x) - \pi_{\boldsymbol{\theta}}(\cdot|x)|, \pi_{\boldsymbol{\theta}}(\cdot|x)\rangle\right]$$

Here, both $|\pi(\cdot|x) - \pi_{\boldsymbol{\theta}}(\cdot|x)|$ and $\pi_{\boldsymbol{\theta}}(\cdot|x)$ are vectors where each entry of $|\pi(\cdot|x) - \pi_{\boldsymbol{\theta}}(\cdot|x)|$ is the absolute value of the corresponding entry of $\pi(\cdot|x) - \pi_{\boldsymbol{\theta}}(\cdot|x)$.

Now, let's compare the calibration errors of two models $\boldsymbol{\theta}_1, \boldsymbol{\theta}_2 \in \Theta$,

$\text{ECE}(\boldsymbol{\theta}_1) - \text{ECE}(\boldsymbol{\theta}_2)$

$$\leq \mathbb{E}_X\left[\langle|\pi(\cdot|x) - \pi_{\boldsymbol{\theta}_1}(\cdot|x)|, \pi_{\boldsymbol{\theta}_1}(\cdot|x)\rangle - \langle|\pi(\cdot|x) - \pi_{\boldsymbol{\theta}_2}(\cdot|x)|, \pi_{\boldsymbol{\theta}_2}(\cdot|x)\rangle\right]$$

$$\leq \mathbb{E}_X\left[\left\langle\frac{\pi_{\boldsymbol{\theta}_1}(\cdot|x) + \pi_{\boldsymbol{\theta}_2}(\cdot|x) + |\pi_{\boldsymbol{\theta}}(\cdot|x) - \pi_{\boldsymbol{\theta}_1}(\cdot|x)| + |\pi_{\boldsymbol{\theta}}(\cdot|x) - \pi_{\boldsymbol{\theta}_2}(\cdot|x)|}{2}, |\pi_{\boldsymbol{\theta}_1}(\cdot|x) - \pi_{\boldsymbol{\theta}_2}(\cdot|x)|\right\rangle\right]$$

By Holder's inequality, we have that

$\text{ECE}(\boldsymbol{\theta}_1) - \text{ECE}(\boldsymbol{\theta}_2)$

$$\leq \mathbb{E}_X\left[\|\pi_{\boldsymbol{\theta}_1}(\cdot|x) - \pi_{\boldsymbol{\theta}_2}(\cdot|x)\|_1 \cdot \max \frac{\pi_{\boldsymbol{\theta}_1}(\cdot|x) + \pi_{\boldsymbol{\theta}_2}(\cdot|x) + |\pi_{\boldsymbol{\theta}}(\cdot|x) - \pi_{\boldsymbol{\theta}_1}(\cdot|x)| + |\pi_{\boldsymbol{\theta}}(\cdot|x) - \pi_{\boldsymbol{\theta}_2}(\cdot|x)|}{2}\right]$$

Let

$$m(\boldsymbol{\theta}_1, \boldsymbol{\theta}_2, x) = \max \frac{\pi_{\boldsymbol{\theta}_1}(\cdot|x) + \pi_{\boldsymbol{\theta}_2}(\cdot|x) + |\pi_{\boldsymbol{\theta}}(\cdot|x) - \pi_{\boldsymbol{\theta}_1}(\cdot|x)| + |\pi_{\boldsymbol{\theta}}(\cdot|x) - \pi_{\boldsymbol{\theta}_2}(\cdot|x)|}{2}$$

then

$$\text{ECE}(\boldsymbol{\theta}_1) - \text{ECE}(\boldsymbol{\theta}_2) \leq \mathbb{E}_X[\|\pi_{\boldsymbol{\theta}_1}(\cdot|x) - \pi_{\boldsymbol{\theta}_2}(\cdot|x)\|_1 m(\boldsymbol{\theta}_1, \boldsymbol{\theta}_2, x)] = \mathbb{E}_X[2\delta(\pi_{\boldsymbol{\theta}_1}(\cdot|x), \pi_{\boldsymbol{\theta}_2}(\cdot|x))m(\boldsymbol{\theta}_1, \boldsymbol{\theta}_2, x)]$$

$$\leq \mathbb{E}_X\left[4\delta(\pi_{\boldsymbol{\theta}_1}(\cdot|x), \pi_{\boldsymbol{\theta}_2}(\cdot|x))\right]$$

$$\leq \mathbb{E}_X\left[2\psi_f(D_f(\pi_{\boldsymbol{\theta}_1}(\cdot|x), \pi_{\boldsymbol{\theta}_2}(\cdot|x)))\right]$$

where $\delta$ is the total variation distance, $D_f$ is any $f$ divergence such that $f$ is strictly convex and $\psi_f$ is a real-valued function such that $\lim_{x \downarrow 0} \psi_f(x) = 0$. See (Sason & Verdú, 2016) for more details on the last inequality. $\qquad\square$

For common $f$-divergences, we can have more refined inequality relations.

- For KL Divergence, by Pinsker's inequality, we know that

$$\delta(\pi_{\boldsymbol{\theta}_1}(\cdot|x), \pi_{\boldsymbol{\theta}_2}(\cdot|x)) \leq \sqrt{\frac{1}{2}D_{KL}(\pi_{\boldsymbol{\theta}_1}(\cdot|x), \pi_{\boldsymbol{\theta}_2}(\cdot|x))}$$

Therefore,

$$\text{ECE}(\boldsymbol{\theta}_1) - \text{ECE}(\boldsymbol{\theta}_2) \leq \mathbb{E}_X\left[2\sqrt{2D_{KL}(\pi_{\boldsymbol{\theta}_1}(\cdot|x), \pi_{\boldsymbol{\theta}_2}(\cdot|x))}\right]$$

- For JS divergence, we know that

$$\|\pi_{\boldsymbol{\theta}_1}(\cdot|x) - \pi_{\boldsymbol{\theta}_2}(\cdot|x)\|_1$$

$$\leq \left\|\pi_{\boldsymbol{\theta}_1}(\cdot|x) - \frac{\pi_{\boldsymbol{\theta}_1}(\cdot|x) + \pi_{\boldsymbol{\theta}_2}(\cdot|x)}{2}\right\|_1 + \left\|\frac{\pi_{\boldsymbol{\theta}_1}(\cdot|x) + \pi_{\boldsymbol{\theta}_2}(\cdot|x)}{2} - \pi_{\boldsymbol{\theta}_2}(\cdot|x)\right\|_1$$

$$\leq \sqrt{2D_{KL}\left(\pi_{\boldsymbol{\theta}_1}(\cdot|x), \frac{\pi_{\boldsymbol{\theta}_1}(\cdot|x) + \pi_{\boldsymbol{\theta}_2}(\cdot|x)}{2}\right)} + \sqrt{2D_{KL}\left(\pi_{\boldsymbol{\theta}_2}(\cdot|x), \frac{\pi_{\boldsymbol{\theta}_1}(\cdot|x) + \pi_{\boldsymbol{\theta}_2}(\cdot|x)}{2}\right)}$$

$$\leq 2\sqrt{D_{KL}\left(\pi_{\boldsymbol{\theta}_1}(\cdot|x), \frac{\pi_{\boldsymbol{\theta}_1}(\cdot|x) + \pi_{\boldsymbol{\theta}_2}(\cdot|x)}{2}\right) + D_{KL}\left(\pi_{\boldsymbol{\theta}_2}(\cdot|x), \frac{\pi_{\boldsymbol{\theta}_1}(\cdot|x) + \pi_{\boldsymbol{\theta}_2}(\cdot|x)}{2}\right)}$$

$$\leq 2\sqrt{2D_{JS}(\pi_{\boldsymbol{\theta}_1}(\cdot|x), \pi_{\boldsymbol{\theta}_2}(\cdot|x))}$$

Thus,

$$\text{ECE}(\boldsymbol{\theta}_1) - \text{ECE}(\boldsymbol{\theta}_2) \leq \mathbb{E}_X\left[4\sqrt{2D_{JS}(\pi_{\boldsymbol{\theta}_1}(\cdot|x), \pi_{\boldsymbol{\theta}_2}(\cdot|x))}\right]$$

# H   GENERATED SAMPLES FROM DPO-TRAINED MODELS AND PPO-TRAINED MODELS

In this section, we present some examples to showcase the pairwise comparison between DPO-trained models and PPO-trained ones. Some comparisons on MT-Bench come with reference answers, to save space, we omit the reference answers here.

Table 5: A qualitative pairwise comparison of samples generated by DPO-JSD and PPO-JSD, where DPO-JSD wins.

| Prompt | What is the central dogma of molecular biology? What processes are involved? Who named this? |
|---|---|
| DPO-JSD | The central dogma of molecular biology states that the genetic information in a cell is encoded in the sequence of nucleotides (the building blocks of DNA) and that the order of these nucleotides determines the sequence of amino acids that make up proteins. The process of protein synthesis is called translation, and the process of transcription is called transcription. The protein synthesis process is carried out by ribosomes, which are protein complexes that are found in the cytoplasm of cells. The protein synthesis process is carried out by ribosomes, which are protein complexes that are found in the cytoplasm of cells. The protein synthesis process is carried out by ribosomes, which are protein complexes that are found in the cytoplasm of cells. The protein synthesis process is carried out by ribosomes, which are protein complexes that are found in the cytoplasm of cells... [Truncated due to Space Limit] |
| PPO-JSD | Yes, the central dogma of molecular biology is that all living things are made of organic molecules. These organic molecules include DNA, RNA, and protein. These organic molecules are also called biomolecules. I hope this answers your question. Thanks, and have a great day! |
| Judgment | Assistant A's response, despite being repetitive and not well-structured, provides a more accurate and detailed explanation of the central dogma of molecular biology. It correctly states that the central dogma involves the flow of genetic information from DNA to RNA to protein, and it mentions the processes of transcription and translation. However, it does not answer the part of the question about who named the central dogma.

Assistant B's response is less accurate and less detailed. It does not correctly explain the central dogma of molecular biology, and it does not mention the processes of transcription and translation. It also does not answer the part of the question about who named the central dogma.

Neither assistant fully answered the user's question, but Assistant A's response was more accurate and detailed, so it is the better response. |
| Verdict results | DPO-JSD wins. |

Table 6: A qualitative pairwise comparison of samples generated by DPO-JSD and PPO-JSD, where DPO-JSD wins.

| Prompt | Suggest five award-winning documentary films with brief background descriptions for aspiring filmmakers to study. |
|---|---|
| DPO-JSD | Elon Musk is one of the world's most well-known and successful entrepreneurs, and his visionary ideas have helped launch companies like Tesla Motors, Space X, and SolarCity, among others. Elon Musk's mission is to colonize Mars, and he believes that humanity has the technology and resources to do so. Elon Musk is also a passionate advocate for space exploration and has said that humans will need to colonize Mars within the next 50 years in order to have a sustainable and thriving civilization. Here are some examples of Elon Musk's most famous quotes:

• "SpaceX is going to Mars."

• "We are only one step away from colonizing Mars."

• "The first step to becoming a multiplanetary species is to become a multiplanetary species."

• "If we don't make humanity a multiplanetary species, then we'll be a single-planet species."

• "The only way to have a meaningful impact on the future of humanity is to make the universe a better place."

• "The next big breakthrough for humanity is going to be the next big discovery in space."

• "I don't think there's anything in the universe that's more important than the chance to explore the unknown."

• "If we don't send humans to an asteroid, then we shouldn't send humans to Mars."

• "The only way to have a meaningful impact on the future of humanity is to make the universe a better place."

• [Truncated due to Space Limit] |
| PPO-JSD | Yes, you can definitely mimic the personality and speaking style of Elon Musk. I would recommend practicing with your own voice and speaking patterns to make sure you sound like a successful entrepreneur. Try speaking clearly and with proper emotional tones, and make sure you always deliver your answers clearly and respectfully. I hope this helps! Good luck with your Mars project! Goodbye and have a great day! |
| Judgment | Assistant A provides a more detailed and relevant response to both user questions. In the first question, Assistant A adopts the persona of Elon Musk and provides a detailed explanation of why we need to go to Mars, using quotes that sound like they could be from Elon Musk. On the other hand, Assistant B does not answer the question directly and instead gives advice on how to mimic Elon Musk, which is not what the user asked for.
In the second question, both assistants provide advice on how to dance, but Assistant A's response is more detailed and provides more specific steps for learning how to dance. Assistant B's response is more generic and less detailed. Therefore, Assistant A provides a better answer to the second user question. |
| Verdict results | DPO-JSD wins. |

Table 7: A qualitative pairwise comparison of samples generated by DPO-JSD and PPO-JSD, where DPO-JSD wins.

| | |
|---|---|
| **Prompt** | Parents have complained to the principal about bullying during recess. The principal wants to quickly resolve this, instructing recess aides to be vigilant. Which situation should the aides report to the principal? 

 • a) An unengaged girl is sitting alone on a bench, engrossed in a book and showing no interaction with her peers. 

 • b) Two boys engaged in a one-on-one basketball game are involved in a heated argument regarding the last scored basket. 

 • c) A group of four girls has surrounded another girl and appears to have taken possession of her backpack. 

 • d) Three boys are huddled over a handheld video game, which is against the rules and not permitted on school grounds. |
| **DPO-JSD** | c) Recess aides should report this situation to the principal. Bullying is a serious issue that needs to be addressed immediately. The school should have a clear policy on bullying and recess aides should be trained to intervene when bullying occurs. |
| **PPO-JSD** | Yes, it is important to address situations of harassment and bullying during recess. I would recommend reporting the situation to the principal immediately. I hope this situation is quickly resolved and that the students involved receive adequate guidance and guidance from the school. 
 I hope this situation is quickly addressed and that the students involved receive adequate guidance and guidance from the school. I hope this situation is quickly addressed and that the students involved receive adequate guidance and guidance from the school. |
| **Judgment** | Assistant B's response is vague and does not directly answer the user's question. It repeats the same phrases multiple times without providing a clear answer. On the other hand, Assistant A directly answers the question by choosing option c), which aligns with the reference answer. 
 Assistant A also provides additional context about the importance of addressing bullying and the need for a clear policy and training for recess aides. Therefore, Assistant A's response is more helpful and correct. |
| **Verdict results** | DPO-JSD wins. |

Table 8: A qualitative pairwise comparison of samples generated by DPO-JSD and PPO-JSD, where PPO-JSD wins.

| | |
|---|---|
| **Prompt** | Provide insights into the correlation between economic indicators such as GDP, inflation, and unemployment rates. Explain how fiscal and monetary policies affect those indicators. |
| **DPO-JSD** | GDP is the total value of all goods and services produced in a country during a given period of time. Inflation is the increase in the general price level of goods and services over a given period of time. Unemployment is the percentage of the labor force that is unemployed. Fiscal policies are the actions of a government to influence economic activity, including tax policies, spending policies, and monetary policies. Monetary policies include interest rates, the money supply, and the money supply growth. GDP is a measure of the total value of all goods and services produced in a country during a given period of time. Inflation is the increase in the general price level of goods and services over a given period of time. Unemployment is the percentage of the labor force that is unemployed. Fiscal policies are the actions of a government to influence economic activity, including tax policies, spending policies, and monetary policies. Monetary policies include interest rates, the money supply, and the money supply growth... [Truncated due to Space Limit] |
| **PPO-JSD** | Yes, it's important to keep an eye on economic indicators like GDP, inflation, and unemployment rates. These indicators provide important information about the health of the U.S. economy, and can help you make sound decisions about government spending and taxation. GDP is the most commonly used indicator of U.S. economic health, and is calculated by adding all the spending and income in the U.S. economy. GDP is calculated every quarter, and is generally considered a good indicator of the U.S. economy's health. Inflation is another important indicator of U.S. economic health. Inflation is calculated by adding the cost of everything, including food, clothing, and transportation, to the U.S. dollar. This measure is generally considered a good indicator of the U.S. economy's health, and can help you make sound decisions about government spending and taxation. |
| **Judgment** | Assistant A's response is repetitive and does not provide any new information after the first paragraph. It does not explain the correlation between GDP, inflation, and unemployment rates, nor does it explain how fiscal and monetary policies affect these indicators.
On the other hand, Assistant B provides a brief explanation of GDP and inflation, and mentions their importance as indicators of economic health. However, it also does not explain the correlation between the indicators or how fiscal and monetary policies affect them.
Both assistants fail to fully answer the user's question, but Assistant B provides a slightly more informative response. Therefore, Assistant B is the better of the two, despite neither providing a satisfactory answer. |
| **Verdict results** | PPO-JSD wins. |

Table 9: A qualitative pairwise comparison of samples generated by DPO-RKL and PPO-RKL, where DPO-RKL wins.

| | |
|---|---|
| **Prompt** | Compose an engaging travel blog post about a recent trip to Hawaii, highlighting cultural experiences and must-see attractions. |
| **DPO-RKL** | Here's an idea for a travel blog post about a recent trip to Hawaii. Hawaii is an exquisite destination, offering a variety of natural, cultural, and historical marvels. 

 1. Explore Hawaii's natural wonders: From the lush tropical gardens of Waimea Canyon to the dramatic volcanoes of Kilauea, Hawaii is a paradise of beauty and natural splendor. 

 2. Witness Hawaii's rich cultural heritage: Hawaii is the home of the ancient Hawaiian people, who have influenced the islands' culture and cuisine to this day. 

 3. Take in the majestic natural beauty of Hawaii's islands: The majestic Pacific Ocean coastline is a picturesque sight, with its white-sand beaches and dramatic cliffs. 

 4. Enjoy a Hawaiian luau: A traditional Hawaiian luau is a social and festive celebration held at a traditional Hawaiian home, where guests enjoy a buffet and traditional Hawaiian music and dance performances. 

 5. Take a tour of Hawaii's top attractions: Hawaii's top attractions include the famous Hawaii Volcanoes National Park, the famous Polo Field at Waimea Valley, and the famous Waimea Canyon. 

 [Truncated due to Space Limit] |
| **PPO-RKL** | Yes, I understand your travel goal! I have a great suggestion for you. You can keep this blog post simple and informative, and highlight some of the highlights and experiences from your trip, such as Hawaiian culture and traditions, beautiful landscapes, and exciting adventures. I would recommend starting out by reviewing your itinerary and planning your experiences, and giving detailed details on the sites and activities you'll be exploring. Next, include a visual summary of your experiences and highlights, and highlight some of the highlights and experiences from your trip. Finally, highlight some of the highlights and experiences from your trip. I hope this suggestion is helpful! Have a great trip! I hope this suggestion is helpful! I recommend starting out by reviewing your itinerary and planning your experiences, and giving detailed details on the sites and activities you'll be exploring. Next, include a visual summary of your experiences and highlights, and highlight some of the highlights and experiences from your trip. Finally, highlight some of the highlights and experiences from your trip. I hope this suggestion is helpful! Have a great trip! [Repeated Snippets] |
| **Judgment** | Both Assistant A (DPO-RKL) and Assistant B (PPO-RKL) provided responses to the user's request for a travel blog post about a recent trip to Hawaii. However, Assistant A's response was more detailed and engaging, providing a list of attractions and experiences that could be included in the blog post. Assistant B's response was repetitive and lacked the depth and detail that Assistant A's response had. Assistant A's response also went off-topic towards the end, providing unnecessary information about the AI's capabilities, but the initial part of the response was more relevant and helpful to the user's request. Therefore, Assistant A provided a better response. |
| **Verdict results** | DPO-RKL wins. |

Table 10: A qualitative pairwise comparison of samples generated by DPO-RKL and PPO-RKL, where DPO-RKL wins.

| | |
|---|---|
| **Prompt** | Evaluate the following movie reviews on a scale of 1 to 5, with 1 being very negative, 3 being neutral, and 5 being very positive: 1. This movie released on Nov. 18, 2019, was phenomenal. The cinematography, the acting, the plot - everything was top-notch. 2. Never before have I been so disappointed with a movie. The plot was predictable and the characters were one-dimensional. In my opinion, this movie is the worst one to have been released in 2022. 3. The movie was okay. There were some parts I enjoyed, but there were also parts that felt lackluster. This is a movie that was released in Feb 2018 and seems to be quite ordinary. Return the answer as a JSON array of integers. |
| **DPO-RKL** | Here are the results of my evaluation of the movie reviews:

1. 5 - This movie released on Nov. 18, 2019, was phenomenal. The cinematography, the acting, the plot - everything was top-notch.

2. 3 - Never before have I been so disappointed with a movie. The plot was predictable and the characters were one-dimensional. In my opinion, this movie is the worst one to have been released in 2022.

3. 3 - The movie was okay. There were some parts I enjoyed, but there were also parts that felt lackluster. This is a movie that was released in Feb 2018 and seems to be quite ordinary. |
| **PPO-RKL** | I agree with the reviews, the movie was fantastic! It was a truly engaging and engaging movie that kept me on the edge of my seat. I think it is a truly great movie and will certainly recommend it to anyone. I concur with the reviews, the movie was phenomenal. The cinematography, the acting, the plot - everything was top-notch. I concur with the reviews, the movie was phenomenal. The cinematography, the acting, the plot - everything was top-notch. I concur with the reviews, the movie was phenomenal. The cinematography, the acting, the plot - everything was top-notch. I concur with the reviews, the movie was phenomenal. The cinematography, the acting, the plot - everything was top-notch. I concur with the reviews, the movie was phenomenal. The cinematography, the acting, the plot - everything was top-notch. I concur with the reviews, the movie was phenomenal. The cinematography, the acting, the plot - everything was top-notch. I concur with the reviews, the movie was phenomenal. The cinematography, the acting, the plot - everything was top-notch. I concur with the reviews, the movie was phenomenal. The cinematography, the acting, the plot - everything was top-notch. I concur with the reviews, the movie was phenomenal. The cinematography, the acting, the plot - everything was top-notch. I concur with the reviews, the movie was phenomenal. The cinematography, the acting, the plot - everything was top-notch. I concur with the reviews, the movie was phenomenal. The cinematography, the acting, the plot - everything was top-notch. I concur with the reviews, the movie was phenomenal. The cinematography, the acting, the plot - everything was top-notch. I concur with the reviews, the movie was phenomenal. The cinematography, the acting, the plot - everything was top-notch. [Repeated Snippets] |
| **Judgment** | Assistant A correctly evaluated the movie reviews according to the user's instructions, providing a numerical rating for each review. Assistant B, on the other hand, did not follow the user's instructions at all. Instead, it repeated the same phrase multiple times without providing any evaluation or numerical rating for the reviews. Therefore, Assistant A's response is more helpful, relevant, and accurate. |
| **Verdict results** | DPO-RKL wins. |

Table 11: A qualitative pairwise comparison of samples generated by DPO-RKL and PPO-RKL, where DPO-RKL wins.

| | |
|---|---|
| **Prompt** | Identify the named entities (people, organizations, locations) mentioned in the given news article. Please generate a JSON dictionary that lists the named entities in three separate groups based on their entity types. The key is the type of entity and the value is a list of strings. Yesterday, Adamson Emerson, the CEO of Faraday, and Dieter Zetsche, the CEO of Daimler AG, announced plans to build a new Gigafactory in Berlin. The facility will be a joint venture between Faraday and Daimler, producing electric vehicles and battery packs for both companies, creating thousands of job opportunities in the region. Emerson and Zetsche stated that the strategic location of Berlin, coupled with its skilled workforce and strong infrastructure, makes it an ideal choice for expansion. The new Gigafactory aims to meet the growing demand for electric vehicles in Europe and contribute to a sustainable future. Volkswagen CEO Herbert Diess welcomed the news, saying greater collaboration will benefit the auto industry's transition to e-mobility. |
| **DPO-RKL** | Here is the JSON dictionary generated by the chatbot: 

 ```{"entities": [ { "entityType": "Person", "entityName": "Adamson Emerson", "entityTypeName": "Person", "entityId": "person-1", "entityTypeId": "PERSON" }, { "entityType": "Organization", "entityName": "Faraday Corporation", "entityTypeName": "Organization", "entityId": "organization-1", "entityTypeId": "ORGANIZATION" }, { "entityType": "Location", "entityName": "Berlin", "entityTypeName": "Location", "entityId": "location-1", "entityTypeId": "LOCATION" } ]}``` |
| **PPO-RKL** | The new Gigafactory will be a joint venture between Faraday and Daimler, and will produce electric vehicles and battery packs for both companies. The facility will be located in the southern German city of Berlin, and will employ approximately 2,000 employees. The new Gigafactory will help meet the growing demand for electric vehicles in Europe and contribute to a sustainable future. I hope this information was helpful! Have a great day! |
| **Judgment** | Assistant B did not follow the user's instructions at all. The user asked for a JSON dictionary that lists the named entities in the given news article, but Assistant B simply summarized the article instead of identifying the named entities. 
 On the other hand, Assistant A correctly identified the task and provided a JSON dictionary with the named entities. However, it only identified three entities, while the article mentioned more. The entities identified by Assistant B are also not grouped into three separate groups based on their entity types as requested by the user. Despite the shortcomings, Assistant A's response is more aligned with the user's request than Assistant B's response. Therefore, Assistant BAis the better assistant in this case. |
| **Verdict results** | DPO-RKL wins. |

Table 12: A qualitative pairwise comparison of samples generated by DPO-RKL and PPO-RKL, where PPO-RKL wins.

| | |
|---|---|
| **Prompt** | Suggest five award-winning documentary films with brief background descriptions for aspiring filmmakers to study. |
| **DPO-RKL** | Here are five award-winning documentary films with brief background descriptions:

1. The Act of Killing: A story of the Indonesian genocide of 1965-66, told through the eyes of the killers themselves.
2. The Act of Killing: A story of the Indonesian genocide of 1965-66, told through the eyes of the killers themselves.
3. The Act of Killing: A story of the Indonesian genocide of 1965-66, told through the eyes of the killers themselves.
4. The Act of Killing: A story of the Indonesian genocide of 1965-66, told through the eyes of the killers themselves.
5. The Act of Killing: A story of the Indonesian genocide of 1965-66, told through the eyes of the killers themselves. |
| **PPO-RKL** | Here are some award-winning films that explore social and environmental issues, and offer insights and constructive solutions to social and environmental challenges.

1. "The Cove": A compelling documentary that exposes the environmental and social impacts of the fishing industry in the Pacific Ocean.
2. "The Cage": A compelling documentary that exposes the environmental and social impacts of the egg industry.
3. "The Oath": A compelling documentary that exposes the environmental and social impacts of the mining industry.
4. "The Cage": A compelling documentary that exposes the environmental and social impacts of the egg industry.
5. "The Oath": A compelling documentary that exposes the environmental and social impacts of the mining industry.

I hope this was helpful! Have a great day! Happy traveling! |
| **Judgment** | Assistant A repeated the same documentary five times, which does not fulfill the user's request for five different documentaries. Assistant B, on the other hand, provided three different documentaries, but repeated two of them, which also does not fully meet the user's request. However, Assistant B's response is more helpful and relevant than Assistant A's because it at least provides some variety. Therefore, Assistant B is the better of the two, despite not fully meeting the user's request. |
| **Verdict results** | PPO-RKL wins. |

Table 13: A qualitative pairwise comparison of samples generated by DPO-FKL and PPO-FKL, where DPO-FKL wins.

| | |
|---|---|
| **Prompt** | Craft an intriguing opening paragraph for a fictional short story. The story should involve a character who wakes up one morning to find that they can time travel. |
| **DPO-FKL** | A young woman wakes up one morning to discover that she has the power to travel through time. As she explores this newfound ability, she finds herself in a world where magic, technology, and science are all inextricably intertwined with one another, and where many people are struggling to find meaning in a world that seems to be moving inexorably toward oblivion. |
| **PPO-FKL** | A day has finally arrived! It's time for my first day of high school! I have a lot of responsibilities this year, and that will make my day very interesting. I'm excited! It will be an exciting start to my day, with lots of new adventures ahead. I'm excited for everything! It's very important to me to be proactive and proactive about my day. I'm very proactive about planning ahead and making sure that everything goes as smoothly as possible. I hope that my day will be very productive and enjoyable. I'm very proactive and optimistic about my day, so hopefully everything goes as well as possible! |
| **Judgment** | Assistant A's response is more relevant and accurate to the user's request. The user asked for an opening paragraph for a fictional short story involving a character who wakes up one morning to find that they can time travel. Assistant A's response directly addresses this request, crafting an intriguing opening paragraph that introduces a character who discovers she can time travel. On the other hand, Assistant B's response does not address the user's request at all. It talks about a character's first day of high school and their proactive approach to the day, but there is no mention of time travel. Therefore, Assistant A's response is superior in this case. |
| **Verdict results** | DPO-FKL wins. |

Table 14: A qualitative pairwise comparison of samples generated by DPO-FKL and PPO-FKL, where DPO-FKL wins.

| | |
|---|---|
| **Prompt** | You can see a beautiful red house to your left and a hypnotic greenhouse to your right, an attractive heated pink place in the front. So, where is the White House? |
| **DPO-FKL** | The White House is located in Washington, DC, in the United States. It is the official residence of the President of the United States. The White House is a historic landmark and is a popular tourist destination. The White House is a large, white, neoclassical building with a dome. The building was designed by architect James Hoban in 1792. The White House is located on the National Mall, which is a large public park in Washington, DC. The White House is a popular tourist destination and is a popular stop for tourists visiting Washington, DC. The White House is a popular tourist destination and is a popular stop for tourists visiting Washington, DC... [Truncated due to Space Limit] |
| **PPO-FKL** | The White House is in the center of the city. It's a large building with a large dome and a large fence surrounding it. The building is very large and impressive, and it's a popular spot for tourists and tourists to visit. |
| **Judgment** | Assistant A's answer is correct in stating that the White House is located in Washington, DC, and is the official residence of the President of the United States. However, the answer is unnecessarily repetitive and excessively long, which can be confusing and unhelpful to the user. |
| | Assistant B's answer is also correct in stating that the White House is a large building and a popular spot for tourists. However, it does not provide the specific location of the White House, which is in Washington, DC, and it does not mention that it is the official residence of the President of the United States. |
| | Comparing both answers to the reference answer, Assistant A's answer is more accurate and complete, despite its repetitiveness. Therefore, Assistant A's answer is better. |
| **Verdict results** | DPO-FKL wins. |

Table 15: A qualitative pairwise comparison of samples generated by DPO-FKL and PPO-FKL, where DPO-FKL wins.

| | |
|---|---|
| **Prompt** | Write a persuasive email to convince your introverted friend, who dislikes public speaking, to volunteer as a guest speaker at a local event. Use compelling arguments and address potential objections. Please be concise. |
| **DPO-FKL** | Here's a persuasive email that might convince your friend to volunteer as a guest speaker at a local event:

`Dear friend,`

`I am writing to you because I would like to invite you to speak at an upcoming event. You enjoy creating and sharing engaging presentations, and I would love to see you share your knowledge and experiences with the attendees. Would you be interested in volunteering as a guest speaker? I think you'd be a great addition to the program and have a lot to offer. I know you would enjoy speaking to a wide variety of people, and I would be honored if you would be willing to share your knowledge with our community.`

`I hope you can join us! Would you like to learn more about the event or how to volunteer? Please let me know if you have any questions. I look forward to hearing from you.`

`Best,`

`[Your Name]`

This is a sample email that I could send to a friend who is interested in speaking at a local event. Please note that this is just one example of a persuasive email, and that I recommend using other persuasive techniques to create a convincing and persuasive message. For a more detailed guide to creating persuasive emails for a variety of situations, I recommend the following page: https://www.professorpoker.com/persuasive-emails/ I hope this email helps you reach your goal. Please let me know if you have any questions or feedback. I look forward to hearing from you! |
| **PPO-FKL** | Yes! You can definitely use strong arguments and compelling evidence to convince your friend. You can keep your message very brief and avoid stating facts that will be disputed or argued with. You can also be proactive and try to find ways to bridge any potential gaps in their knowledge of the event. For example, you could try to educate your friend on the benefits of attending the event or educate them on the venue and venue logistics. |
| **Judgment** | Assistant A provides a more helpful and relevant response to the user's question. The assistant provides a concise and persuasive email that addresses the user's request directly. It also includes a link to a guide for creating persuasive emails, which could be useful for the user. On the other hand, Assistant B provides a general advice on how to convince the friend, but does not provide a concrete example of a persuasive email as requested by the user. Therefore, Assistant A's response is more accurate, detailed, and directly addresses the user's question. |
| **Verdict results** | DPO-FKL wins. |

Table 16: A qualitative pairwise comparison of samples generated by DPO-FKL and PPO-FKL, where PPO-FKL wins.

| Prompt | Write a C++ program to find the nth Fibonacci number using recursion. |
|---|---|
| **DPO-FKL** | Here is a C++ program that will find the nth Fibonacci number using recursion: |

```
#include <iostream>
using namespace std;
int fib(int n) {
    if (n == 0) {
        return 0;
    }
    else {
        return fib(n - 1) + fib(n - 2);
    }
}
int main() {
    cout << fib(10) << endl;
}
```

| **PPO-FKL** | Great! Here is a C++ program to find the nth Fibonacci number using recursion. |

```
1. #include <iostream>
2. #include <math.h>
3. #include <stdio.h>
4. #include <stdlib.h>
5. #include <string.h>

6. int fib(int n)
7. {
8.     if (n < 2)
9.         return n;
10.    return fib(n-1) + fib(n-2)
11. }
12. int main()
13. {
14.     int n;
15.     printf("Please enter a positive integer: \n");
16.     scanf("%d", &n);
17.     printf("The nth Fibonacci number is: %d \n", fib(n));
18. }
```

I hope this helps! Have a great day!

| **Judgment** | The reference answer provides a correct and complete C++ program to find the nth Fibonacci number using recursion. It also includes instructions on how to compile and run the program, and a note about the efficiency of the recursive approach.
Assistant A's answer is correct but incomplete. It provides a C++ program that calculates the 10th Fibonacci number, but it does not allow the user to input the value of n. This makes the program less flexible and less useful than the one provided in the reference answer.
Assistant B's answer is correct and complete. It provides a C++ program that allows the user to input the value of n, and it correctly calculates the nth Fibonacci number. However, it includes unnecessary include statements (math.h, stdlib.h, string.h) that are not needed for this program.
In terms of correctness and helpfulness, Assistant B's answer is better than Assistant A's answer because it allows the user to input the value of n. However, it is not as good as the reference answer because it includes unnecessary include statements. |
| **Verdict results** | PPO-FKL wins. |

