# OpenReview forum: "Beyond Reverse KL: Generalizing Direct Preference Optimization with Diverse Divergence Constraints"
_ICLR.cc/2024/Conference — ICLR 2024 spotlight_

### Official Review · Reviewer_sLFR · 2023-10-30

**Soundness:** 3 good
**Presentation:** 2 fair
**Contribution:** 2 fair
**Rating:** 5
**Confidence:** 3

**Summary:**

This paper proposes a general method that utilizes diverse divergence constraints and direct preference optimization to achieve alignment with human preferences. This paper shows that the reward can be reparameterized using the policy model and a reference model. Namely, it is possible to solve the RLHF problem via a supervised learning approach under a broad class of divergence constraints. The divergence regularization can be used to trade-off between accuracy and diversity. The experiment results show that the proposed method is more stable and efficient in optimization than the RL method.

**Strengths:**

> This paper investigates the AI alignment performance under the constraint of many popular divergences and analyzes the effectiveness of these regularizations. This introduces greater flexibility for the fine-tuning process using human preference.

> This paper establishes the relationship between the reward function and the optimal policy. It proves that the language model can be optimized using the policy model and the reference model with divergence constraints. This method is a supervised learning approach and sounds more stable than RLHF.

> The empirical results show that the balance between the alignment accuracy and the diversity can be adjusted using different divergence regularizations. In addition, the proposed framework shows comparable (and, in some cases, better) performance and greater divergence efficiency than PPO-based methods.

**Weaknesses:**

> The novelty of the method is limited. DPO uses the reverse KL regularization constraint to achieve the mapping between the reward model and the optimal policy. The proposed method only extends the reverse KL regularization into a broad class of commonly used divergences.

> There is still room to innovate on the theoretical findings. The derivations in the method section take up much space. However, they are mainly from DPO - the difference is that the authors introduce the Lagrange multiplier for the constrained (but in practice, we do not need to deal with these constraints because they are naturally satisfied) and use Df rather than DKL.

**Questions:**

> What are the assumptions and limitations of f-DPO, and how do they affect its applicability?

> What is the advantage of using different divergences compared to tuning DPO’s beta hyperparameter to balance the accuracy and diversity?

---

> ### Author Response · Authors · 2023-11-15
>
> We deeply appreciate the reviewer for the thoughtful comments! You can find our responses to each question below.
>
>
> 1. **Re weakness 1 (novelty)**:
>
> Thank you for the comments. In our work, we build upon the formulation introduced in the DPO paper, which analytically links the reward function to the policy. A critical aspect to note is that the DPO's derivation, while comprehensive, **only** facilitates a closed-form mapping between the policy and the reward model for reverse KL divergence. This limitation highlights the need for a more nuanced analysis, particularly when considering other types of divergences. Extending from reverse KL to other divergences is not merely a matter of substituting the divergence constraint, as the original derivation of DPO will not lead to a closed form mapping for other divergences.
>
> Our work steps in to fill this gap by meticulously examining the KKT conditions, rather than directly following the DPO's derivation. This approach has enabled us to demonstrate that a closed-form mapping between the policy and reward model is indeed possible for a variety of commonly used divergences. This finding represents an important and non-trivial advancement beyond the original scope of the DPO paper.
>
> To further address the reviewer's concern about the novelty of our method: While it is true that our work extends the concept of reverse KL regularization to include a wider array of divergences, the implications of this extension are profound. The inclusion of multiple divergences not only broadens the applicability of the model but also introduces extra flexibility and adaptability in policy optimization. This improvement is useful in scenarios where different divergences may yield more optimal results, depending on the specific characteristics and requirements of the task at hand. Therefore, our contribution should be viewed not merely as an extension, but as an enrichment of the existing DPO framework.
>
>
>  2. **Re weakness 2 (theoretical findings)**:
>
> Thank you for the suggestion on improving the theoretical findings! We recognize the importance of emphasizing the novel aspects of our theoretical contributions. While our derivations are built on the foundation laid by DPO, our approach differs in several aspects. Primarily, our method extends the applicability of closed-form mappings to a broader range of divergences, not just the reverse KL divergence. This extension is non-trivial and addresses a significant gap in the DPO paper.  Furthermore, our work introduces a detailed exploration of the relationship between various f-divergences and the model's calibration error as well as a theoretical bound. This aspect of our research explains why divergence efficient algorithms should be favored.
>
> Regarding the transition from RKL to f-divergences and the use of the Lagrange multiplier in our derivations, we would like to note that these changes are not simply replacements of divergences. Merely applying the derivation from the original DPO paper does not yield a closed-form solution for the functional mapping between policy and reward. Therefore, we respectfully disagree with the reviewer that the derivations are mainly from the DPO paper.

---

> ### Author Response · Authors · 2023-11-15
>
> 3. **Re Q1 (assumptions and limitations):**
>
> Thank you for the good question. In our f-DPO framework, the key assumptions are relatively minimal and generally satisfied in standard language models. The reviewer can also find those assumptions in section 4, page 6. We restate the assumptions below. Firstly, we assume that the base language model assigns a non-zero prediction probability to all tokens, a condition typically met by models employing softmax functions. This ensures that our framework is applicable to a wide range of existing language models.
>
> Regarding the divergence function  $f(\cdot)$ used in $f$-DPO, we require three main conditions:
>   - 1) $ 0 \notin \text{dom}(f') $ (zero is not in the domain of the derivative of $ f$ );
>   - 2) $ f(1) = 0 $ (the function evaluates to zero at 1), and that
>   - 3) $ f $ is strictly convex around 1.
>
> These conditions are met by several commonly used divergences, including $\alpha$-divergence, Reverse KL Divergence (RKL), Forward KL Divergence (FKL), and Jensen-Shannon Divergence (JSD). Thus, our model is adaptable to a variety of divergence choices.
>
> However, it's crucial to also consider the limitations and their implications for $f$-DPO's applicability. The constraints on the divergence function $f$ may limit the selection of divergences to those that conform to these specific conditions, potentially excluding some divergences (e.g., total variation distance, Chi-squared distance) that might offer unique benefits in certain applications.
>
>
> 4. **Re Q2 (tuning $\beta$):**
>
> This is a really good observation! Adjusting the $\beta$ hyperparameter can influence the diversity, but it will also influences the regularization of divergence from the base model! A higher value of  beta can enhance generation diversity, but may lead to poor performance on the downstream task. On the other hand, using different divergence measures allows for a more nuanced control over this balance. Different divergences can offer alternative ways of quantifying and constraining the deviation, potentially allowing for increased diversity in outputs while still maintaining closer alignment with the original model's accuracy and reliability. This approach can provide a more flexible and targeted method to balance accuracy and diversity, compared to solely adjusting the beta hyperparameter.
>
> In essence, the advantage of using different divergences lies in their ability to provide a more tailored approach to balancing model accuracy with output diversity, potentially leading to more effective and controlled modifications than simply tuning the beta value.
>
>
> ----
>
> Once again, we appreciate your time and constructive feedback. If further questions or concerns arise, please don't hesitate to reach out. Thank you!

---

### Official Review · Reviewer_AH62 · 2023-11-01

**Soundness:** 3 good
**Presentation:** 4 excellent
**Contribution:** 3 good
**Rating:** 8
**Confidence:** 2

**Summary:**

This paper takes a look at recent RLHF works that use policy gradient methods in conjunction with divergence-to-the-initial-model regularizations, and through some derivations show that they all belong in a similar family of methods. These equivalences lead the authors to propose a generalized form of DPO, a purely supervised objective, which is derived by combining the general f-divergence based family of methods with a Bradley-Terry type reward.

The authors show that using the proposed method on RLHF datasets allows for faster training on the reward and better calibrated models than PPO counterparts, and show that the choice of f-divergence induces interesting trade offs.

**Strengths:**

The paper is well written, and in my book is good science: generalizing from several methods, taking care to do proper evaluation of the performance _and_ the behaviors that may lead to the performance, as well as an evaluation of the different trade offs offered by the method.

I'm not sure I'll be the best judge of impact but this seems like a set of valuable insights into the space of RLHF methods.

**Weaknesses:**

I don't have much bad to say about the paper, although I am wondering if there could not be a more complete set of baselines (see below).

**Questions:**

It feels like there's a missing baseline (although this may be a criticism of RLHF more than of this specific work), which is an off-policy offline RL baseline, such as CQL [1] and recent variants. As the authors correctly point out, training a model with (on-policy) policy gradient methods such as PPO require rollouts, whereas purely supervised methods such as that proposed in the paper do not. This creates an evaluation mismatch though. It may be the the real difference between f-DPO and PPO is not the different use of divergence per-se but rather the pure use of offline data. As far as I can tell the present work does not account for that.

[1] Conservative Q-Learning for Offline Reinforcement Learning, Aviral Kumar, Aurick Zhou, George Tucker, Sergey Levine, 2020

---

> ### Author Response · Authors · 2023-11-15
>
> We deeply appreciate the reviewer for the thoughtful comments! You can find our responses to each question below.
>
>
> 1. **Re weakness and questions:**
>
> Thank you to the reviewer for bringing up an important topic!
>
> In our work, both $f$-PPO and $f$-DPO optimize the policy under the same objective, as stated in Section 4, Page 5. The primary difference lies in their methodologies.
>   - PPO adopts a two-stage method: it initially learns a reward function from offline data and then optimizes the policy using policy gradient techniques under the learned reward model.
>   - In contrast, DPO establishes a closed-form functional relationship between the policy and the reward function, allowing us to optimize the policy in a supervised manner using offline data.
>
> Theoretically, both methods should converge to the same solution. However, in practice, due to the inherent instability of RL training – as also pointed out in the original DPO paper – PPO tends to underperform compared to DPO. In our experiments, we observe that incorporating the divergence penalty into the reward function exacerbates this instability. The varying value of the divergence, particularly for forward KL divergence, complicates the learning of the value function and makes gradient estimation more error-prone. Thus, the enhanced performance of f-DPO is largely attributable to its more stable training strategy.
>
>
> Additionally, as noted by Gao et al. (2023), reward hacking presents a significant challenge in RLHF, often due to imperfect proxies or limited offline data. While offline RL shows promise in addressing reward hacking in policy optimization, both PPO and $f$-DPO, despite utilizing offline data, do not effectively handle this issue. We typically resort to early stopping as a mitigation strategy. Therefore, we believe that incorporating design principles from offline RL algorithms, such as CQL or IQL, could improve the DPO framework by preventing reward hacking in a more principled way. In this context, offline RL algorithms could be seen as complementary to both PPO and $f$-DPO.  However, we do agree with the reviewer that adding more baseline will further strengthen the findings of our work. In the future, we will consider incorporating design principles from offline RL (e.g., pessimism) into the $f$-DPO framework to further enhance its robustness against reward hacking.
>
> ----
>
> Once again, we appreciate your time and constructive feedback. If further questions or concerns arise, please don't hesitate to reach out. Thank you!
>
>
> **References:**
>
> Gao, Leo, John Schulman, and Jacob Hilton. "Scaling laws for reward model overoptimization." In International Conference on Machine Learning, pp. 10835-10866. PMLR, 2023.

---

> > ### Comment · Reviewer_AH62 · 2023-11-22
> >
> > That's all good to know, thanks for the response.

---

### Official Review · Reviewer_GUwk · 2023-11-03

**Soundness:** 3 good
**Presentation:** 3 good
**Contribution:** 3 good
**Rating:** 8
**Confidence:** 4

**Summary:**

This paper generalises the direct preference optimisation (DPO) framework to a broad class of divergences, i.e., f-divergence. With the similar ideas of DPO, f-DPO uses the reward model, BT model, and the KKT conditions to deal with the constraints. Experiments demonstrates that the proposed f-DPO outperforms the two PPO-based baselines.

**Strengths:**

DPO is an elegant framework for RLHF. This paper is an important extension of DPO to a more general class. The writing of the paper is clear and the experiments are sufficient.

**Weaknesses:**

I think one issue is that how to select the f-divergence. Given that there are many options of the divergence, how to select the best one. Other questions can be found in the questions section.

**Questions:**

There are several questions I want the authors to address during the rebuttal:
1. how to select the f-divergence? The proposed framework is general, that is great. However, when applying to the real scenarios, how to select the best f-divergence? From the experiments, the FKL and RKL can be the options. But more discussion can be included. Is it possible to optimise the f-divergence simultaneously with the policy?

2. One observation is that RKL will harm the diversity and FKL can do better on the diversity. Is there any explanation about this? I think instead of showing the experiments, a more empirical or theoretical analysis would be helpful for us to understand the necessities of f-divergence.

I generally think this is a good paper. So not many questions are raised.

---

> ### Author Response · Authors · 2023-11-15
>
> Many thanks to the reviewer for the thoughtful observations and comments! Our detailed responses to the questions can be found below.
>
>
> 1. **Re Q1 and Weakness:**
>
> Thanks for the great question. In response to the question on selecting the appropriate f-divergence for f-dpo, we suggest initially considering the Jensen-Shannon divergence. Our recommendation is based on its ability to yield more diverse responses compared to the commonly used reverse KL divergence. This preference is supported by our findings, where JS divergence can result in a more diversified response as measured by different diversity metrics. Additionally, despite underperforming in reverse KL in reward metrics, it was favored in the GPT-4 evaluation, as illustrated in Figure 7 of the appendix. This observation also aligns with recent studies, such as Sun et al., 2023.
>
> Furthermore, to accommodate diverse application needs, another strategy could involve training multiple models, each employing a different divergence, such as JS divergence, alpha divergence (with alpha values ranging from 0.1 to 0.9), and forward KL divergence. This approach allows users to experiment and choose the divergence that best aligns with their specific requirements or preferences.
>
> Additionally, while our current framework does not optimize f-divergence simultaneously with the policy, this poses an intriguing area for future research. Incorporating simultaneous optimization could potentially further enhance the adaptability and effectiveness of the framework in various real-world scenarios.
>
> To address your concerns, we’ve also added the above discussions in our revised paper. You can find them in the Section C of Appendix. The revisions are marked in red color.
>
>
> 2. **Re Q2 (RKL vs FKL):**
>
> The distinction between Reverse KL Divergence (RKL) and Forward KL Divergence (FKL) primarily lies in their inherent properties: RKL is mode-seeking, while FKL is mass-covering. This can be discerned from their mathematical formulations. RKL is defined as $ E_q[\log(q(x)/p(x))] $, where $ q(x) $ is the distribution being optimized, typically representing the language model undergoing fine-tuning. In the RKL scenario, $ q(x) $ may assign zero probability to values where $ p(x) > 0 $. In contrast, FKL is defined as $ E_p[\log(p(x)/q(x))] $, which necessitates that $ q(x) $ assigns a non-zero probability to all values where $ p(x) > 0 $. This requirement inherently encourages FKL to promote a distribution $ q(x) $ that covers the entire range of $ p(x) $, thereby enhancing diversity.
>
> As for the reviewer's inquiry regarding the impact of RKL and FKL on diversity, the observed differences can be attributed to these underlying properties. RKL's mode-seeking behavior tends to focus on specific peaks in the probability distribution, potentially leading to less diverse outcomes. Conversely, FKL's mass-covering nature ensures a broader coverage of the probability landscape, contributing to greater diversity in the results. This theoretical perspective, supplemented by empirical evidence as shown in our experiments, underscores the importance of carefully selecting the appropriate f-divergence based on the desired characteristics of the language model.
>
> The above discussions are included in Section D of appendix in our revised version, please feel free to check them out!
>
>
> ----
>
> Once again, we appreciate your time and constructive feedback. If further questions or concerns arise, please don't hesitate to reach out. Thank you!
>
> **References**:
>
> Sun, Simeng, Dhawal Gupta, and Mohit Iyyer. "Exploring the impact of low-rank adaptation on the performance, efficiency, and regularization of RLHF." arXiv preprint arXiv:2309.09055(2023).

---

> > ### Comment · Reviewer_GUwk · 2023-11-21
> > **Thanks for the response**
> >
> > Thanks for the detailed response. I think the response addresses my concerns and will keep my score.

---

### Official Review · Reviewer_Lpy5 · 2023-11-06

**Soundness:** 4 excellent
**Presentation:** 3 good
**Contribution:** 4 excellent
**Rating:** 8
**Confidence:** 3

**Summary:**

This paper extends the Direct Preference Optimization (DPO) algorithm used for training LLMs with RLHF to handle different $f$-divergences between distributions, which have differing tradeoffs between alignment and diversity. The current DPO approach optimizes the reverse KL divergence between the learned and reference model, however the reverse KL is widely known to induce mode-collapsing behavior, which explains the loss of diversity with increased alignment. This paper gives a closed form solution for a variety of $f$ divergence, including reverse KL, forward KL, Jenson-Shannon and different $\alpha$-divergences. Each of these have different tradeoffs between mode-seeking and mode-covering behaviors (reverse KL is the most mode-seeking, forward KL mode-covering). They evaluate language models fine-tuned with these different divergences across 3 datasets (IMDB, Anthropic HH and MT-Bench). They find that training with the different divergences indeed induce different tradeoffs between reward maximization and diverse generations. The existing DPO algorithm, which uses reverse KL, maximizes the reward but is the least diverse; forward KL is the most diverse but lowest reward, and different $\alpha$ divergences are in the middle. The experiments also show that for the same divergence, $f$-DPO consistently outperforms the PPO analog in terms of generation quality as evaluated by GPT4. Finally, a nice experiment which is in the appendix shows that GPT4 also judges DPO with some diversity (like using the JSD) over DPO with reverse KL, which is consistent with the intuition that somewhat diverse text is more realistic.

Overall, I think this is a nice paper that should be accepted. It proposes a clean, theoretically grounded and computationally efficient solution for handling the tradeoff between alignment and diversity for finetuning LLMs with RL, which I think will be of broad interest. The paper is also very nicely written. While the experiments are on a somewhat small scale in terms of LLM and datasets size, I think they are sufficient to confirm the behavior predicted by the theory, namely that optimizing the different divergences induces different tradeoffs between alignment and diversity, and these trends are consistent across the different experiments. This is sufficient for a first paper, and I expect that the community will extend these evaluations to larger models and datasets.

**Strengths:**

- Simple, effective and potentially very useful method for inducing differing tradeoffs between alignment and diversity when finetuning LLM with preference models. I think this is timely and will be of broad interest to the community working on fine-tuning LLMs with RL.
- Well-written paper
- Nicely designed experiment section

**Weaknesses:**

- Experiments are on a somewhat small scale with older LLMs - ideally it would have been nice to see experiments with more modern LLMs like LLaMA-7B. However, I think the method is principled enough and the trends in the experiments robust enough that I am optimistic the results will transfer to larger models.

**Questions:**

My main suggestion for improving the paper would be evaluating the approach on more modern LLMs like LLaMA. I understand these experiments can be computationally expensive though, and I don't think they are essential for the paper to be accepted.

---

> ### Author Response · Authors · 2023-11-15
>
> We appreciate the reviewer for the thoughtful comments!
>
> 1. **Re weakness and questions:**
>
> We agree that incorporating experiments with larger models, such as LLaMA 7B, would indeed strengthen our work. We chose the Pythia 2.8B model in alignment with the DPO paper. Should resources allow, we plan to rerun the experiments using the LLaMA 7B model and include the results in the final version!
>
> ----
>
> Thank you again for your valuable time and consideration. Should you have any additional concerns or questions, kindly let us know. Thank you!

---

### Meta-Review · Area_Chair_6syU · 2023-12-05

**Metareview:**

The current paper extends Direct Preference Optimization (DPO) for AI alignment of LLMs, where the reverse KL (RKL) divergence is replaced by a broader family i.e. Csiszar f-divergence. The replacement is justified by analyzing the KKT conditions of the associated optimization. The resulting f-DPO gives a trade-off between alignment performance and generation diversity and outperforms the original DPO on RLHF datasets.

Strengths:
- Simple yet meaningful generalization of DPO to tradeoff between alignment and diversity
- Experiments are well designed and support the authors' claim on better efficiency and divergence.
- This well-written work meets well with the community's interest in LLMs.

Weaknesses:
- Experiments are on relatively small-scale datasets.

The authors are highly recommended to revise the paper based on the reviewers' comments. Besides, please find some additional remarks below.

In sec 4, it is not clear what reasons led to the adoption of the f-divergence (e.g. based on the specific formula of the f-div, or its specific properties). Note that the f-divergence is a very large family, while the experiments are mainly on Jensen-Shannon divergence and alpha-divergence (including FKL and RKL as limit cases when alpha=0 or 1, this should also be mentioned).

During the discussion, some reviewer suggested showing examples of the benefits of different 'f' in *specific* applications to help the reader understand the advantage of f-DPO.

**Justification For Why Not Higher Score:**

There is still some place of improvement on larger-scale LLMs, and on the selection/adoption of f-divergences. The introduction of f-divergence can be improved.

**Justification For Why Not Lower Score:**

This work demonstrates both theoretical (adoption of advanced information theoretical tools) and practical strengths (proper evaluation on LLMs) and aligns with the community interest.

---

### Decision · Program_Chairs · 2024-01-16

Accept (spotlight)